# HOPS/CORVET tethering complexes are critical for endocytosis and protein trafficking to invasion related organelles in malaria parasites

Joëlle Paolo Mesén-Ramírez[1,2,3], Gwendolin Fuchs[1,2,3¤], Jonas Burmester[1,2,3], Guilherme B. Farias[1,2,3], Ana María Alape-Flores[4], Shamit Singla[5,6], Arne Alder[1,2,3], José Cubillán-Marín[2], Carolina Castro-Peña[2], Sarah Lemcke[1,3], Holger Sondermann[1,7,8], Mónica Prado[4], Tobias Spielmann[2], Danny Wilson[1,2,5,6], Tim-Wolf Gilberger[1,2,3]*

1 Centre for Structural Systems Biology, Hamburg, Germany, 2 Bernhard Nocht Institute for Tropical Medicine, Hamburg, Germany, 3 University of Hamburg, Hamburg, Germany, 4 Microbiology Faculty and Center for Research in Tropical Diseases (CIET), University of Costa Rica, San José, Costa Rica, 5 Research Centre for Infectious Diseases, School of Biological Sciences, University of Adelaide, Adelaide, Australia, 6 Institute for Photonics and Advanced Sensing (IPAS), University of Adelaide, Adelaide, Australia, 7 Deutsches Elektronen-Synchrotron DESY, Hamburg, Germany, 8 Christian-Albrechts-Universität zu Kiel, Kiel, Germany

¤ Current Address: Instituto Gulbenkian de Ciência, Oeiras, Portugal
* gilberger@bnitm.de

## Abstract

The tethering complexes HOPS/CORVET are central for vesicular fusion through the eukaryotic endolysosomal system, but the functions of these complexes in the intracellular development of malaria parasites are still unknown. Here we show that the HOPS/CORVET core subunits are critical for the intracellular proliferation of the malaria parasite *Plasmodium falciparum.* We demonstrate that HOPS/CORVET are required for parasite endocytosis and host cell cytosol uptake, as early functional depletion of the complex led to developmental arrest and accumulation of endosomes that failed to fuse to the digestive vacuole membrane. Late depletion of the core HOPS/CORVET subunits led to a severe defect in merozoite invasion as a result of the mistargeting of proteins destined to the apical secretory organelles, the rhoptries and micronemes. Ultrastructure-expansion microscopy revealed a reduced rhoptry volume and the accumulation of numerous vesicles in HOPS/CORVET deficient schizonts, further supporting a role of HOPS/CORVET in post-Golgi protein cargo trafficking to the invasion related organelles. Hence, malaria parasites have repurposed HOPS/CORVET to perform dual functions across the intraerythrocytic cycle, consistent with a canonical endocytic pathway for delivery of host cell material to the digestive vacuole in trophozoite stages and a parasite specific role in trafficking of protein cargo to the apical organelles required for invasion in schizont stages.

## Author summary

Malaria pathology is caused by the proliferation of *Plasmodium* parasites within red blood cells. HOPS/CORVET are tethering complexes critical for vesicle fusion and cargo

**Data availability statement:** All relevant data are within the manuscript and its Supporting Information files.

**Funding:** This work was supported by the Deutsche Forschungsgemeinschaft (DFG, German Research Foundation; DFG - Deutsche Forschungsgemeinschaft) – Project-number 453548970, GRK 2771 (to TWG) and by the Leibniz ScienceCampus InterACt (https://www.leibniz-interact.de/en/) funded by the BWFGB Hamburg and the Leibniz Association - Project-number W75/2022 InterACt (to TWG), the Alexander Von Humboldt Fellowship (to DW) (Humboldt Research Fellowship - Alexander von Humboldt-Foundation)-Award number 3.1-AUS-1221926-HFST-E, the Australian Research Council RTP scholarship (to SS) Research Training Program - Department of Education, Australian Government and a DAAD/ Universities Australia Collaborative Research Grant - Project-number 5770226 (to DW and TWG). GBF received his salary from funding by the DFG (GRK 2771). The funders had no role in study design, data collection and analysis, decision to publish or preparation of the manuscript.

**Competing interests:** The authors have declared that no competing interests exist.

transport in eukaryotes but the role of these complexes in the development of *Plasmodium falciparum* parasites remains unclear. Using a conditional system to deplete the core subunits of HOPS/CORVET at different time points we identified essential functions for these complexes. *Plasmodium* parasites internalize hemoglobin from the host cell cytosol and digest it in a digestive vacuole (DV). Functional depletion of HOPS/ CORVET leads to the accumulation of host cell cytosol containing vesicles, indicating an important role of HOPS/CORVET in the fusion of these vesicles to the DV membrane. Merozoites, the invasive stage of malaria parasites, have specific organelles, the rhoptries and micronemes, which contain and secrete proteins necessary for the invasion of new red blood cells. HOPS/CORVET deficient parasites showed a severe invasion defect and an aberrant localization of proteins destined to these organelles, indicating a critical role of these complexes for the vesicular transport and cargo delivery to the rhoptries and micronemes. Our study pinpoints a stage-specific dual function of HOPS/CORVET for endocytosis and for vesicular transport to invasion related organelles.

## Introduction

Malaria is the most important parasitic disease worldwide with an estimated 249 million cases leading to more than 600 000 deaths in 2022 [1]. The disease is caused by apicomplexan parasites of the genus *Plasmodium*. Of the five human infecting species *Plasmodium falciparum* causes the most severe form of the disease and is responsible for >90% of the deaths [1,2]. Despite significant advances in the control of the disease, since 2020 the estimated number of malaria-related deaths has been on the rise, due in part to disruptions in malaria control caused by the COVID-19 pandemic [3]. The pathology of the disease is caused by the asexual stages of the parasite which proliferate within human red blood cells (RBC).

The asexual blood cycle begins when a merozoite invades an erythrocyte, forming a parasitophorous vacuole (PV) [4] within which the small single nuclei ring stage parasite is established. To grow and replicate within the RBC, the parasite needs to endocytose and degrade more than 80% of the host cell cytosol [5] that is composed of 95–98% hemoglobin [6]. This parasite-driven endocytic pathway transports hemoglobin via vesicular trafficking to a lysosomal-like organelle termed "digestive vacuole" where hemoglobin is finally metabolized down to amino acids and the heme group is detoxified by bio-crystallization into hemozoin [7]. As the parasite grows it undertakes several cycles of mitotic replication, becoming a syncytial schizont stage. This is followed by cytokinesis and segmentation into up to 32 daughter merozoites [8]. Merozoites are highly polarized cells and possess phylum specific secretory organelles termed rhoptries, micronemes and dense granules. These organelles are generated *de novo* in every asexual replication cycle and secrete proteins that are critical for host cell invasion [9]. The intraerythrocytic development of the parasite culminates with the rupture of the infected RBC and the release of the invasive merozoites which then invade new RBCs.

In eukaryotes, the majority of endocytosed or secreted proteins transit through a series of membrane-bound organelles via transport vesicles. During endocytosis, these vesicles can dynamically interconvert and are comprised of early endosomes, recycling endosomes, late endosomes and lysosomes – referred to together as the endolysosomal system [10]. For secretion, the Golgi apparatus acts as an initial hub for the budding and sorting of vesicles destined to secrete their contents into another organelle or outside the plasma membrane [11]. The vesicular trafficking system´s dynamic nature requires the coordinated functions

of multiple molecular mechanisms like cargo selection, vesicle budding, maturation and vesicle fusion [12]. Some of these basic mechanisms are widely conserved in eukaryotes and the proteins involved show functional homology for orthologues across species. Intracellular membrane fusion systems, for example, require a canonical machinery [13] that consists of i) Rab GTPases, which initiate the fusion process and once activated recruit, ii) multisubunit tethering complexes, which bring into contact two membranes that contain, iii) membrane-anchored receptor proteins (SNAREs) which enhance the membrane contact and mediate the actual membrane fusion process [14,15].

Membrane tethering provides a physical link between two membranes that will be fused and is one of the earliest steps, before SNARE complex formation, that define the specificity of membrane fusion [16,17]. The tethering complexes CORVET (class C Core vacuole/endosome tethering) [18] and HOPS (homotypic fusion and vacuole protein sorting) [19,20] are multisubunit complexes critical for the tethering of endocytic and exocytic Golgi-derived vesicles along the endolysosomal system of eukaryotes [21]. In yeast and mammalian cells, both complexes comprise an identical 4 subunit core consisting of the Vacuolar Protein Sorting (VPS) proteins 11, 16, 18 and 33. Each complex has additional specific subunits that likely define the specificity of membrane interaction and fusion by interacting with activated Rab proteins and paired SNAREs on membrane surfaces [13] (Fig 1A). In the canonical CORVET and HOPS model, CORVET associates with early endosomes as they fuse with and mature to late endosomes, with this pathway directed by the specific subunits VPS3 and VPS8 which interact with Rab5. Whereas HOPS associates with late endosomes from the endocytic pathway or vesicles transporting proteases from the Golgi to the lysosome, in a pathway directed by the specific subunits VPS39 and VPS41 and featuring important interactions with Rab7 and AP-3 [22] for cargo delivery.

The composition of the core subunits is generally considered to be conserved among eukaryotes but species-specific variations in both composition and function have been described for CORVET and HOPS complexes [13,16,23]. In the case of the apicomplexans, growing evidence supports that these parasites have repurposed some components of the classical endocytic pathway to the secretory pathway [24] such as Rab11a [25], VPS45 [26], DrpB [27] and sortilin [28], with both pathways intersecting at the *trans* Golgi [29] forming an endosomal- like compartment (ELC), a sorting platform for proteins trafficking to the secretory organelles [30]. Additionally, apicomplexan parasites appear to have lost or simplified tethering complexes. One example is the exocyst, a multisubunit protein complex that mediates the tethering of secretory vesicles to the plasma membrane [31] which is greatly reduced in apicomplexans [32,33]. Another example is the Dsl1 complex, that recognizes Golgi-derived COPI vesicles and delivers them over to the fusion machinery at the ER membrane [34] which is absent in apicomplexans [32].

In *Toxoplasma gondii*, HOPS and CORVET have been identified and the core subunits were reported to play essential roles in the biogenesis of secretory organelles and in downstream events such as invasion and gliding motility [23,35]. Surprisingly, given the importance of HOPS and CORVET for endocytosis in most other organisms, there is currently no evidence that these complexes have a role in this process in *T. gondii*, although endocytosis has been reported to have a role in plasma membrane homeostasis and nutrient acquisition [36,37]. In contrast, endocytosis of host-cell material is known to be critical for malaria parasite survival [5,38,39]. Furthermore, proteins (and membrane lipids) of the secretory organelles are thought to be Golgi-derived and trafficked via the endolysosomal pathway [40]. However, whether HOPS and CORVET have a role in the tethering and sorting machinery that controls endocytosis of host-cell material and the formation of the secretory organelles in malaria parasites has yet to be determined.

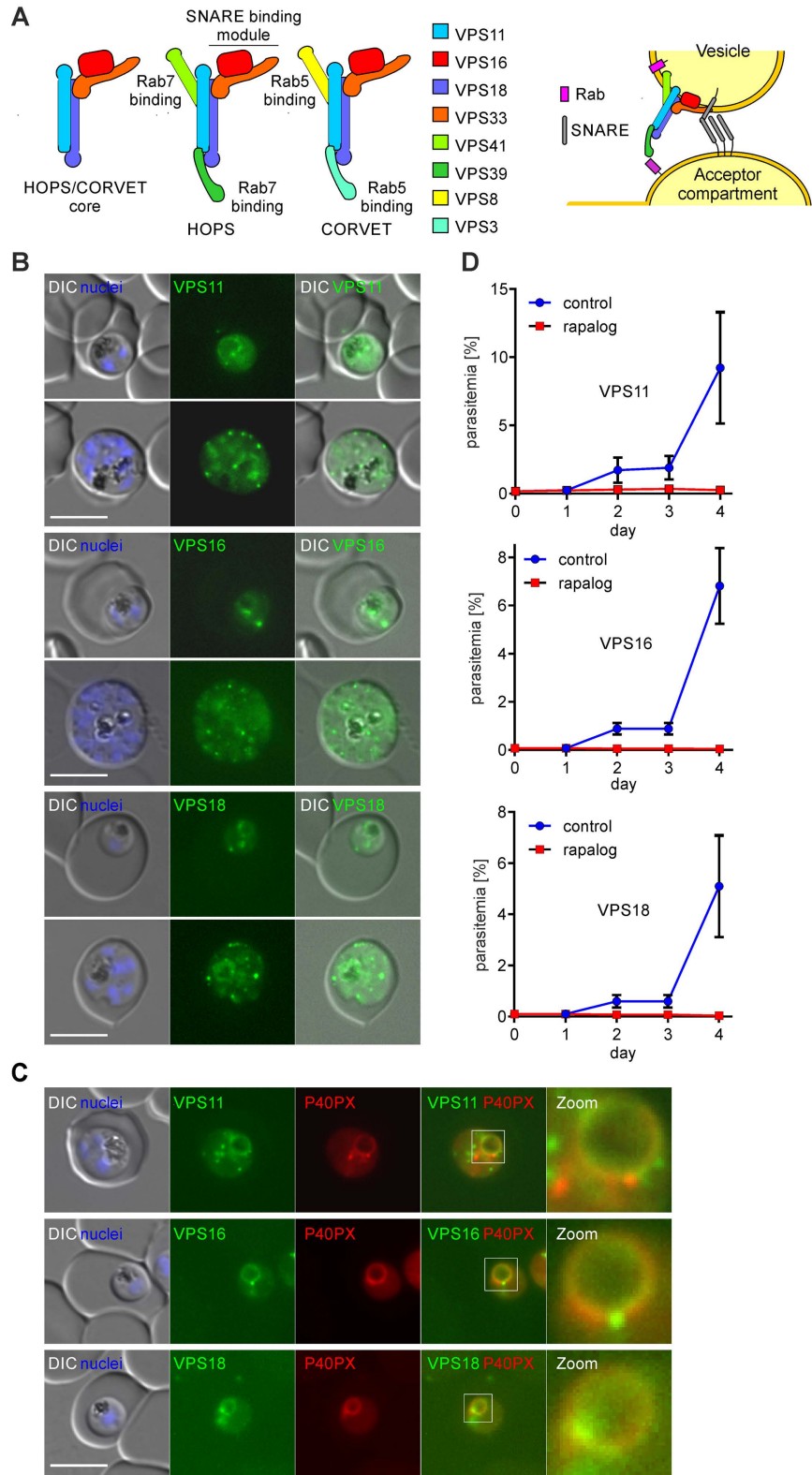

**Fig 1. The HOPS/CORVET core subunits are important for intracellular replication. (A)** Left, schematic representation of the composition of the HOPS and CORVET complexes modified from the predicted architecture in yeast [41,42]. Right, model of HOPS/CORVET -mediated tethering of a vesicle and an acceptor compartment

upon interaction with the corresponding Rab and SNARE proteins before fusion. Modified from [41]. **(B)** Live-cell fluorescence microscopy images of transgenic parasite lines expressing the indicated 2xFKBP-GFP-2xFKBP tagged endogenous VPS11, 16 and 18. **(C)** Live cell fluorescence microscopy images of transgenic parasite lines expressing endogenous VPS11/16/18-2xFKBP-GFP together with P40PX-mCherry as marker for endosomes and the digestive vacuole membrane. Nuclei were stained with DAPI. DIC, differential interference contrast. Scale bars: 5 μm. Enlarged micrographs (zoom: 650x) of the indicated white boxes are shown to visualise localization of the VPS subunit in relation to the P40PX (DV membrane). **(D)** Flow cytometry growth curves over four days of the indicated cell lines in absence (control) or presence of rapalog. Dots indicate mean parasitemia of n = 4 replicates and error bars indicate SD.

Here, using a conditional knock-sideways approach we pinpoint the function of the core subunits of the tethering complexes HOPS/CORVET in the malaria parasite *P. falciparum*. We show that the HOPS/CORVET core tethering subunits play a dual role in the intracellular proliferation of malaria parasites since these complexes are critical for i) endocytosis of host cellular material and ii) the trafficking of proteins to the apical secretory organelles that are essential for the invasion of new red blood cells. These new insights demonstrate that the HOPS/CORVET complex can be repurposed to have dual functions across different stages of cell cycle development.

## Results

### Inducible knock-sideways of HOPS/CORVET subunits leads to severe growth defects during early and late stages of parasite development

To identify key proteins for vesicular tethering in *P. falciparum*, we performed a comparative genomic analysis against CORVET and HOPS components described in *Saccharomyces cerevisiae* [41,42], *Homo sapiens* [16], *T. gondii* [23,35] and metazoans [43,44] (S1 Table). Proteins previously identified in *T. gondii* [23,35] were used as initial queries to BLAST for potential homologues in the malaria parasite. The *P. falciparum* genome encodes the four common core components of the tethering complex consisting of VPS11, VPS16, VPS18 and VPS33 (Fig 1A and S1 Table). The predicted *T. gondii* genome has an annotated VPS39 but not VPS3 while the *P. falciparum* genome has an annotated CORVET specific subunit VPS3 (PF3D7_1423800). When the annotated *T. gondii* VPS39 was searched against the *P. falciparum* genome, the closest match was the currently annotated *Pfvps3* (S1 Table). Homologues for the CORVET and HOPS specific subunits VPS8 and VPS41 are not currently annotated in the malaria genome, but BLAST analysis using the VPS 41/8 domain of *Tgvps8* as query identified a conserved gene with unknown function (PF3D7_0916400) sharing 48% identity. This suggests that apicomplexans have some differences from the canonical organization described in yeast and mammalian cells and, similarly to *T. gondii*, *P. falciparum* parasites appear to lack some of the typical accessory subunits linked to HOPS and CORVET in model cell systems.

Next, we attempted to modify the endogenous locus of the four core VPS subunits and the specific CORVET subunit VPS3 in *P. falciparum* via selection linked integration (SLI) [45] to fuse 2xFKBP-GFP-2xFKBP to the C-terminus of each subunit (S1A and S1B Fig). The tagged subunits (VPS3, 11, 16 and 18) were detectable, albeit lowly expressed, from the trophozoite stage and showed GFP foci, resembling vesicle-like structures, that increased in number as parasites matured and the number of nuclei increased (Figs 1B and S1C). In trophozoites, a GFP signal surrounding the digestive vacuole (DV) was observed for all tagged subunits which was confirmed through co-localization using the DV membrane-located mCherry-tagged Phosphatidylinositol 3-phosphate (PI3P) binding protein P40PX as a fluorescent reporter [39,46] (Fig 1C). GFP foci for the tagged core subunits were also found to lie proximally to the Golgi apparatus using GRASP as reporter (S1D Fig) and to the mScarlet-tagged reporters

AP-3 [47,48], Rab6 [49] and Rab7 [50], which localize to endolysosomal compartments (S1E Fig). Attempts to tag the fourth core subunit VPS33 with GFP and the smaller HA-tag were unsuccessful, suggesting that C-terminal tagging is detrimental for protein function.

To dissect the function of the tethering complexes we targeted the individual subunits using the knock-sideways (KS) system, a conditional approach suited for rapid functional depletion of trafficking factors via the FRB-FKBP dimerization system [45,51,52]. In this system, the subunit of interest is fused to a 2xFKBP-GFP-2xFKBP tag while the FRB is fused to a nuclear mislocalizer. Addition of a small ligand (rapalog) induces the dimerization and recruitment (mislocalization) of the target subunit to the nucleus away from its site of function and is expected to functionally deplete the protein of interest. We modified the pSLI integrated 2xFKBP-GFP-2xFKBP plasmid to also co-express the nuclear mislocalizer (*nmd3*'1xNLS-FRB) from a second promoter [53] in the resulting integrant cell lines (VPS3/11/16/18-2xFKBP-GFP-2xFKBP) (S1A Fig). Recruitment of proteins to the nucleus using this method is not in itself detrimental for parasite development [39,45,54], hence any growth phenotype observed is expected to be largely the result of the removal of the VPS subunit of interest from its cellular compartment.

To analyse the importance of the VPS11, VPS16 and VPS18 HOPS/CORVET core subunits for blood stage development, we monitored the parasitemia of synchronised cultures grown without (control) or in presence of rapalog over two replication cycles (4 days) using flow cytometry. Mislocalization (from now on functional depletion or knock-sideways (KS)) of each core VPS subunit (VPS11/16/18-KS) caused a pronounced growth defect (VPS11: $97.19 \pm 2.65\%$; VPS16: $99.33 \pm 0.26\%$; VPS18: $99.33 \pm 0.45\%$) in comparison to the control parasites with no increase in parasitemia after the first replication cycle (Figs 1D and S1F). These results indicate an essential role of all targeted core HOPS/CORVET subunits in asexual blood stage proliferation of *P. falciparum* parasites. In contrast, the VPS3 subunit seems to play a minor role in the function of the tethering complex and parasite development, as VPS3 KS led only to a growth reduction ($41.30 \pm 7.30\%$) after two replication cycles (S1F Fig). Due to this limited growth defect, further characterisation of HOPS/CORVET components was focussed on the three core subunits VPS11, VPS16 and VPS18. To assess if this minor growth defect compared to the core subunits was due to differences in knock-sideways efficacy, we counted the number of cells with nuclear (full mislocalization), vesicular (no mislocalization) or partial mislocalization for each subunit. Although some individual parasites displayed a partial mislocalization (strong nuclear localization and some vesicular foci), the subunit of interest was fully mislocalized to the nucleus in the majority of the parasites (VPS3: $75.4 \pm 4.8\%$; VPS11: $59.2 \pm 31.5\%$; VPS16: $57.5 \pm 5.1\%$; VPS18: $68.8 \pm 6.4\%$) upon rapalog induction. The system showed no apparent differences between the lines in the efficacy to mislocalize the corresponding subunit (S2A-S2E Fig).

To further analyze the phenotypical outcomes of the KS of the HOPS/CORVET subunits and define at which point of the intracellular cycle these parasites arrest, parasites were synchronized to a time window of 4 h and i) stages were analysed from Giemsa smears at different time points (Figs 2A and S3A-S3C), ii) cell size was measured (Fig 2B) and iii) number of nuclei in late parasite stage were counted (Fig 2C) in control and KS-induced cultures.

VPS11-KS and VPS18-KS induced in early ring stages (0–4 h.p.i) led to a delay in development from the trophozoite stage onwards with parasites arrested as aberrant trophozoites that did not complete schizogony (Figs 2A and S3A, S3C). These trophozoites showed vesicular structures in the cytosol (Fig 2A) and were visibly smaller in size from the trophozoite stage onwards (26–34 h.p.i). This size reduction became more evident at late stages and increasingly pronounced in VPS18-KS parasites (Fig 2A and 2B).

To compare the number of nuclei in control and KS- induced parasites, parasites were treated with Compound 2 (C2) [55] to prevent the rupture of schizont stage parasites and

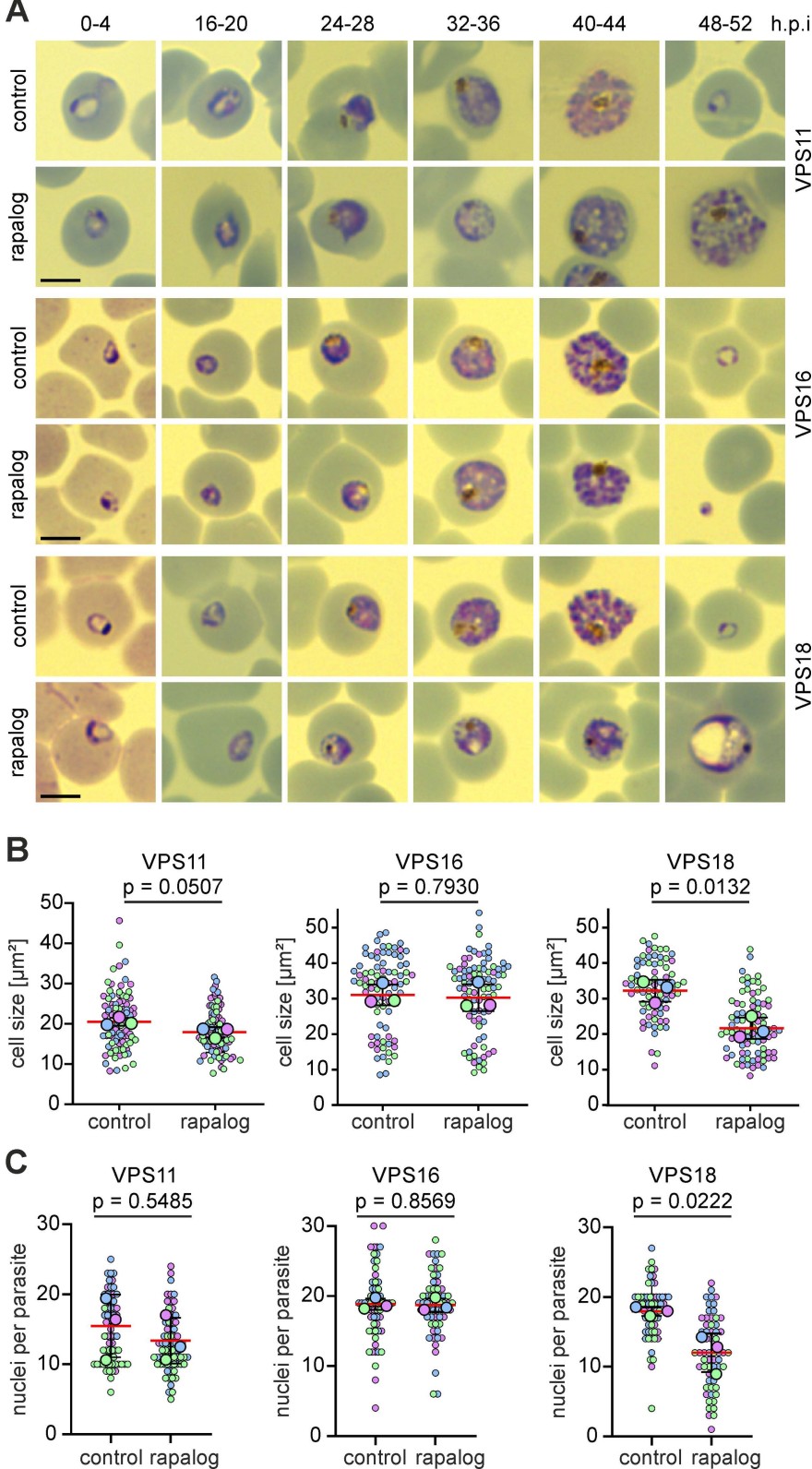

**Fig 2. Stage specific functional depletion of HOPS/CORVET leads to developmental arrest in different stages.**
**(A)** Giemsa-smears images of the indicated parasite cell lines at the indicated time points in absence (control) or presence of rapalog when KS was induced 0–4 h.p.i. **(B)** Cell size (μm²) of control and KS-induced (rapalog) late stages

(36–48 h.p.i) of the indicated cell lines after early induction. Superplot from n = 3 independent experiments and a total of 88 (control) and 95 (rapalog) VPS11 parasites; 79 (control) and 87 (rapalog) VPS16 parasites; 74 (control) and 86 (rapalog) VPS18 parasites. Small dots represent parasites from n = 3 individual experiments defined by blue, purple and green. Large dots, average of each experiment. Mean, red line, error bars (SD), black lines. p values from a two-tailed unpaired t-test are indicated. **(C)** Number of nuclei in control and KS-induced parasites (rapalog) at 36-48 h.p.i when KS was induced early (0–4 h.p.i). Superplot from n = 3 independent experiments and a total of 66 (control) and 72 (rapalog) VPS11 parasites; 59 (control) and 60 (rapalog) VPS16 parasites; 60 (control) and 60 (rapalog) VPS18 parasites. Small dots represent parasites from n = 3 individual experiments defined by blue, purple and green. Large dots, average of each experiment. Mean, red line, error bars (SD), black lines. p values from a two-tailed unpaired t-test are indicated.

ensure parasites were of the same age despite the apparent growth delay after target functional depletion. Quantitation of the number of nuclei (Fig 2C) demonstrated that VPS18-KS (12.11 ± 2.8), but not VPS11-KS (13.37 ± 3.3) parasites exhibited a reduced number of nuclei per parasite compared to controls (VPS18, 18.02 ± 0.65; VPS11, 15.47 ± 4.5). However, late stage schizonts in VPS11-KS parasites appeared to have aberrant or incomplete merozoite formation after this early KS induction (Fig 2A). Both KS parasite lines also revealed a decreased ring stage parasitemia (VPS11, 80.15 ± 11.90%; VPS18, 86.52 ± 6.52%) in the next cycle (S3D Fig), supporting that there is a defect in merozoite development in VPS-11 KS parasites even as the number of nuclei is not directly impacted. These data indicate that early functional depletion of VPS18 and VPS11 leads to a developmental arrest that impacts normal schizont development and subsequent replication.

In contrast, KS induction of VPS16 parasites in early ring stage (0–4 h.p.i) had no apparent effect on the stage and pace of parasite development (Fig 2A, middle row) or the relative size of the parasite compared to control parasites (Figs 2B and S3B). To confirm that VPS16-KS parasites were able to grow and multiply normally, we counted again the number of nuclei (stained with DAPI) of C2- treated parasites at a late time point (46–50 h.p.i) and found no difference in the number of nuclei per schizont (18.52 ± 5.69 (control) versus 18.72 ± 4.45 nuclei per parasite (rapalog)) (Fig 2C). However, a drastic reduction in ring stage parasitemia (88.44 ± 3.56%) was observed after one replication cycle (Figs 1D and S3D). These data indicated that the most significant impact of VPS16-KS, in contrast to VPS11-KS and VPS18-KS, was between rupture of schizonts and merozoite invasion into new RBCs.

## HOPS/CORVET core subunits VPS11 and VPS18 are required for host cell cytosol endocytosis

Since we observed that VPS11 and VPS18 have a role through the early stages of intracellular development and their functional depletion led to accumulation of vesicles (Figs 2A, S4A and S4B), mimicking phenotypes observed when endolysosomal factors are ablated [39,53,54], we proceeded to investigate the role of the HOPS/CORVET core subunits in host cell cytosol uptake (HCCU) in *P. falciparum*. To characterize the function of HOPS/CORVET in HCCU, we performed the bloated digestive vacuole assay where synchronized parasites are grown in the presence of E64, a protease inhibitor that prevents hemoglobin proteolysis in the DV, leading to accumulation of undigested hemoglobin and swelling of the DV (bloating) if hemoglobin trafficking to the DV is properly occurring [39]. When HCCU is defective, the DV will not bloat, as limited hemoglobin is trafficked to the final destination. Quantification of the number of cells with bloated and non-bloated DVs in control and KS-induced (rapalog) cultures showed an ~80% reduction in the number of trophozoites (26–34 h.p.i) with a bloated DV when VPS11 (14.2 ±7.7%) and VPS18 (12.4 ±6.7%) (Fig 3A) were mislocalized compared to control parasites, indicating a defective hemoglobin trafficking to the DV upon KS of

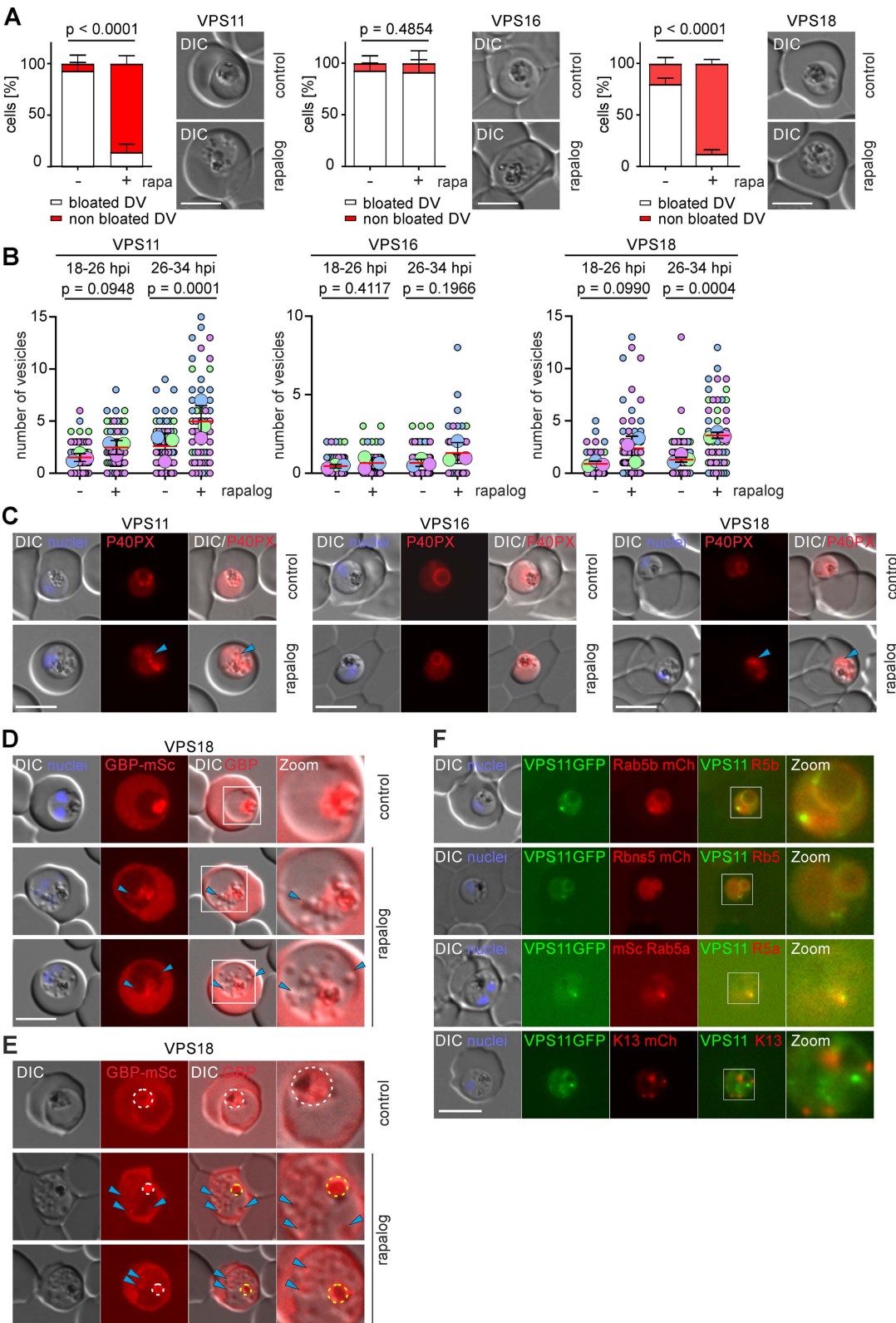

**Fig 3. Functional depletion of VPS11 and VPS18 but not VPS16 causes an endocytosis defect. (A)** Bloated vacuole assay. Left: quantification of the number of cells with bloated and non-bloated DVs in control (-rapa) and KS-induced (+rapa) cultures. Results from n = 3 independent experiments with a total of 136 (control) and 123 (rapalog) VPS11 parasites; 136

(control) and 126 (rapalog) VPS16 parasites; 88 (control) and 88 (rapalog) VPS18 parasites. p values from a Fisher's exact test are indicated. Right, representative live cell images of control and KS-induced parasites (rapalog) scored in a bloated vacuole assay. DIC, differential interference contrast. Scale bars: 5 μm. **(B)** Quantification of number of vesicles in synchronous trophozoites at the indicated time points after KS induction (+ rapalog) compared to controls (-). Superplots from n=3 independent experiments with a total of 78 (control) and 66 (rapalog) VPS11 parasites, 59 (control) and 66 (rapalog) VPS16 parasites, 65 (control) and 62 (rapalog) VPS18 parasites at 18-26 h.p.i; 73 (control) and 60 (rapalog) VPS11 parasites, 67 (control) and 60 (rapalog) VPS16 parasites, 70 (control) and 50 (rapalog) VPS18 parasites at 26-34 h.p.i. Small dots represent parasites from n = 3 experiments defined by blue, purple and green. Large dots, average of each experiment. Mean, red line, error bar (SD), black lines. p values from a two-tailed unpaired t-test are indicated. **(C)** Representative live cell fluorescence images of the indicated parasite cell lines expressing P40PX-mCherry after early KS induction (rapalog) compared to control parasites. Blue arrows indicate PI3P positive vesicles. **(D, E)** Live cell fluorescence images of the VPS18-2xFKBP-GFP parasites expressing an exported protein (GBP$^{1-108}$-mScarlet) after early KS induction (rapalog) compared to control parasites. Enlarged micrographs (zoom: 600x) of the indicated white boxes are shown to visualise overlap of mScarlet signal with the vesicle-like structures visible in the DIC (blue arrows). In (E) parasites were incubated with E64. Bloated or non-bloated DV (pointed circle) is showed. **(F)** Live cell fluorescence images of the VPS11-2xFKBP-GFP parasites co-expressing Rab5b-mCherry, Rbns5-mCherry, mScarlet-Rab5a and Kelch13 mCherry. Enlarged micrographs (zoom: 650x) of the indicated white boxes are shown to visualise localization of the VPS11 in relation to the endolysosomal marker. DIC, differential interference contrast. Scale bars: 5 μm.

VPS11 and VPS18 function. In accordance, the size of the hemozoin was significantly reduced in VPS11-KS parasites (S4C Fig). In contrast, most of the trophozoites with mislocalized VPS16 (Fig 3A, middle row) displayed a bloated DV (91.4% ± 11.9%) comparable to the controls (92.9% ± 7.2%, Fig 3A), suggesting that HCCU is not negatively affected in VPS16-KS parasites despite efficient functional depletion of the protein via KS (S2D Fig).

We noticed from Giemsa smears and live cell microscopy images that VPS11-KS and VPS18-KS parasites exhibited an accumulation of cytosolic vesicle-like structures similar to previously reported phenotypes after inactivation of HCCU by the KS of VPS45 [39], Rbns5 and Rab5b [53] (S4A and S4B Fig). Hence, we quantified the number of vesicles per parasite in early trophozoites (18–26 hpi) and late trophozoites (26–34 hpi). VPS11-KS (Fig 3B left) and VPS18-KS (Fig 3B right) parasites had a significant build-up of vesicles in the cytosol, apparent at early trophozoites (18–26 hpi) and this became more pronounced at late trophozoite stages (26–34 h.p.i), suggestive of a defect in trafficking of HCC containing vesicles to the DV. In contrast, we observed no accumulation of vesicles in VPS16-KS parasites in early and late trophozoite stages comparable to control parasites (Fig 3B middle).

To confirm that these vesicle-like structures are endosomal intermediates involved in the trafficking of hemoglobin to the DV, we used the P40PX m-Cherry reporter [39] (Fig 1C) which labels PI3P positive membranes (early endosomes and DV membrane). We observed that the vesicles in VPS11-KS and VPS18-KS parasites were labelled by P40PX-mCherry compared to control parasites that showed only localization of P40PX at the DV membrane (Figs 3C and S4D). These vesicles were found mostly adjacent to the DV (64% of the vesicles in VPS11- KS and 71% in VPS18-KS) (Fig S4D). This effect was not observed in VPS16-KS parasites, which showed no increase in the number of vesicles and P40PX was found only at the DV membrane (Fig 3C).

To confirm that these vesicles are endosomes containing host cell cytosol that have failed to fuse to the DV and not the result of other secondary effects (e.g., fragmentation of the DV or defects in membrane recycling), we expressed in VPS18 parasites a mScarlet reporter exported to the HCC (GBP130, glycophorin binding protein 130, 1–108 aa), which - once endocytosed - should be contained in those vesicles (Fig 3D) and can be visualized microscopically. Consistently, in VPS18-KS-parasites the mScarlet signal within the parasite overlapped with the vesicles confirming that these structures are endosomes derived from the HCCU pathway (Figs 3D and S4E). Treatment of these parasites with E64 confirmed the lack of DV swelling

(Fig 3A) as a result of the accumulation of vesicles containing HCC that did not reach the DV (Fig 3E).

Since VPS11 and VPS18 were strongly implicated in endolysosomal trafficking, we next sought to determine whether these core HOPS/CORVET subunits co-localized with known markers of the parasite's endolysosomal pathway. In the canonical model of the yeast HOPS/CORVET complex, CORVET interacts with Rab5 proteins in early plasma membrane derived endosomes [18]. To test whether the core malaria parasite HOPS/CORVET complex might function in early endosomes, we co-localized VPS11 with Rbns5 and Rab5b, proteins known to be critical for the delivery of endocytosed hemoglobin via early endosomes to the DV [53]. Rbns5 and Rab5b-mCherry both co-localized with VPS11 at the DV membrane and vesicle-like foci. Rab5a, which has been reported to not be involved in endocytosis [45,53], appeared proximal to the vesicular foci of VPS11 but not at the DV membrane. Kelch13, which localizes to the cytostomes during the early stages of endocytosis before HCC-filled vesicles are generated [54], did not co-localize with VPS11 (Fig 3F). These data support that the core HOPS/CORVET complex in malaria parasites is directly involved in endocytosis of host-cell hemoglobin, likely by influencing the transport of HCC-filled vesicular structures to the DV or fusion with the DV membrane.

## Functional depletion of HOPS/CORVET core subunits leads to a severe defect in merozoite invasion

In *T. gondii*, the HOPS/CORVET core subunits VPS11 and VPS18 have been shown to be necessary for protein trafficking to, and biogenesis of, the invasion related organelles (e.g., rhoptries, micronemes) and for all downstream events needed for invasion of new host cells [23,35]. Our results with VPS16 suggest that this protein might be involved in late-stage development and invasion of new red blood cells. Due to the early developmental arrest and effect on endocytosis caused by functional depletion of VPS11 and VPS18, and the expected involvement of VPS11, VPS16 and VPS18 in the core HOPS/CORVET complex, we proceeded to determine if later KS (before segmentation, 32–36 h.p.i) had a specific impact on schizont development, parasite egress or merozoite invasion. When KS was induced late in the cycle, we observed no apparent morphological difference between control and KS-schizonts (Fig 4A) and no significant difference in the number of nuclei per schizont (Fig 4B), but cultures exhibited a reduction in ring stage parasitemia in the following cycle that was comparable to the growth defect seen with early induction (S3D Fig). These data suggested that late functional depletion of VPS11 and VPS18 impacted on merozoite invasion and initiation of the next cycle of growth.

To confirm that late VPS11, VPS16 and VPS18 functional depletion had no detrimental effect in the segmentation of daughter cells, we expressed IMC1c-mCherry (an inner membrane complex (IMC) protein) marker in these cell lines. VPS11-KS, VPS16-KS and VPS18-KS parasites all successfully established their IMC in mature schizonts upon late VPS11, 16 and 18 KS induction (Figs 4C-E), indicating that nascent merozoites segregate properly upon late KS and that HOPS/CORVET are not required for the formation of the IMC.

After establishing that late functional depletion of VPS11, VPS16 and VPS18 allowed for completion of schizogony, but still no new rings were detected in culture, we proceeded to distinguish if this phenotype is caused by a defect in merozoite egress or merozoite invasion or both. For this, control and KS parasites (KS induced 32–36 h.p.i) were arrested on C2, washed and egress and invasion were allowed to proceed for 4 hours. Ring and schizont parasitemia was then quantified by flow cytometry and from Giemsa-stained smears before and after C2-removal. We observed no significant difference in the egress rate of control and KS parasites for all subunits (Fig 4F). Conversely, we observed a significant reduction in the invasion

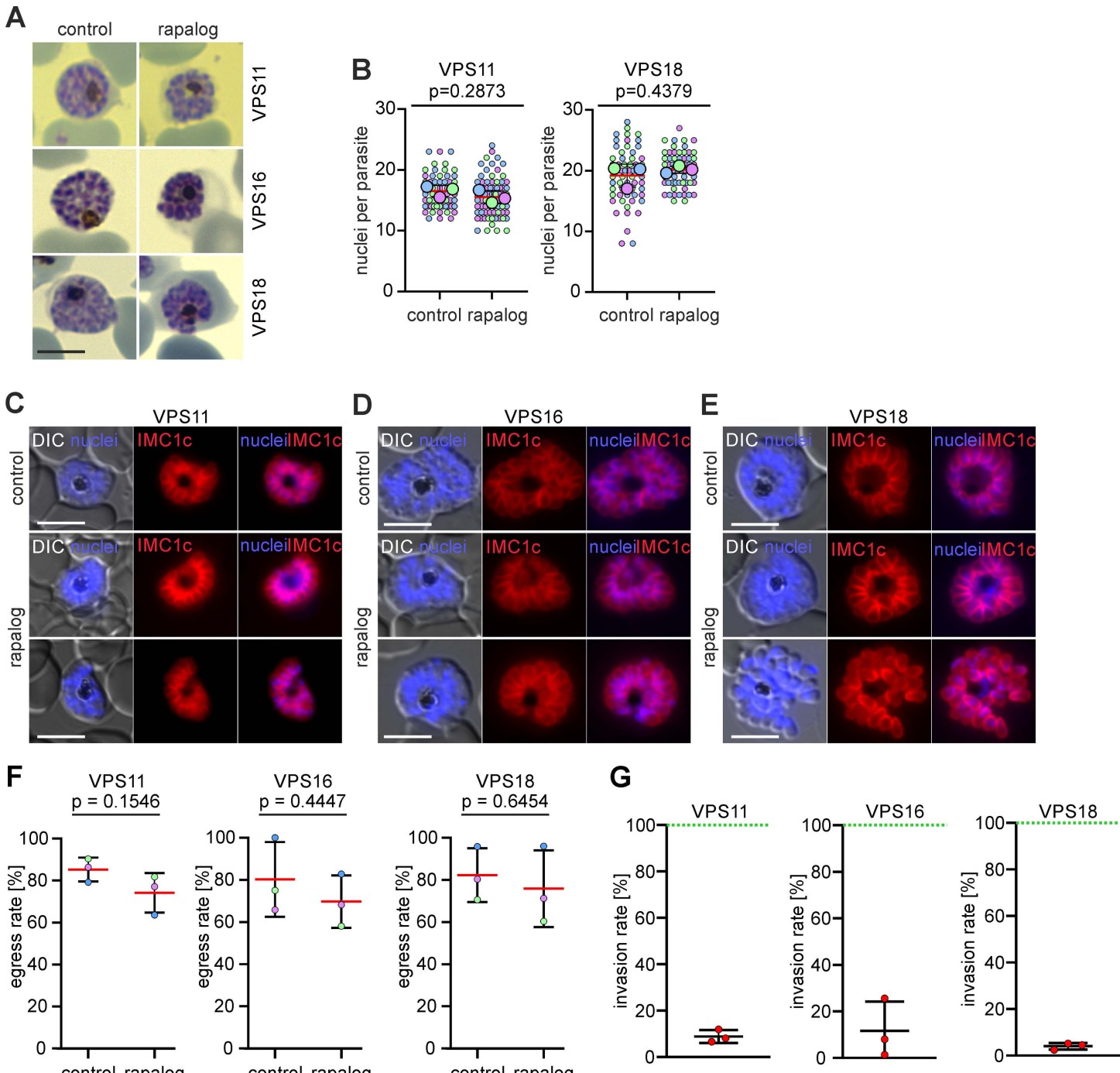

**Fig 4. Knock sideways of HOPS/CORVET subunits leads to an invasion defect. (A)** Giemsa smear images of schizont stages of the indicated cell lines in absence (control) or presence of rapalog when KS was induced in late stages. Scale bars: 3 μm. **(B)** Number of nuclei in control and KS-induced parasites (rapalog) at 48-52 h.p.i when KS was induced late (32-36 h.p.i.). Superplot from n = 3 independent experiments and a total of 58 (control) and 76 (rapalog) VPS11 parasites; 60 (control) and 61 (rapalog) VPS18 parasites. Small dots represent parasites from individual experiments defined by blue, purple and green. Large dots, average of each experiment. Mean, red line, error bar (SD), black lines. p values from a two-tailed unpaired t-test are indicated. **(C-E)** Live cell fluorescence microscopy images of C2- arrested control and late KS-induced (rapalog) schizont stages expressing IMC1c-mCherry. Nuclei were stained with DAPI. DIC, differential interference contrast. Scale bars: 5 μm **(F)** Egress rate of the indicated lines calculated from the number of control and KS-induced schizonts before and after removal of C2. Results from n=3 independent replicates defined by colours. Mean, red line; error bars, SD. p values from a two-tailed unpaired t-test are indicated. **(G)** Invasion rate of the indicated lines calculated from the number of rings per ruptured schizont upon removal of C2. Values from KS-induced parasites (rapalog) were plotted versus values of control parasites (set as 100%, green line).

rate (number of rings per ruptured schizont) of VPS11-KS (−91.11 ± 2.7%), VPS16-KS (−88.3 ± 12.5%) and VPS18-KS (−95.95 ± 1.4%) compared to controls (Fig 4G). These data show that the lack of new rings observed upon late KS is caused by a defect in merozoite invasion rather than a defect in egress.

## HOPS/CORVET core subunits are critical for protein trafficking to the rhoptries

After confirming that functional depletion of VPS11, VPS16 and VPS18 leads to a significant reduction in merozoite invasion, we next assessed whether this was due to a defect in formation of the apical organelles, the rhoptries and micronemes. Rhoptries and micronemes are formed *de novo* at the end of each cycle of blood stage parasite development [56] and they store and secrete proteins essential for the different steps of invasion. To investigate if the cause for the observed invasion phenotype is a defect in organelle biogenesis and/or protein trafficking to the organelles, we generated parasite cell lines co-expressing mCherry-tagged RON12 (a luminal soluble rhoptry neck protein) [57] (Fig 5A-C) and ARO (a rhoptry bulb protein associated to the cytosolic side of the rhoptry membrane) [58] (Fig 5E-G) to visualize the effect of the KS in protein trafficking to the rhoptries. C2-arrested control schizonts expressing RON12-mCherry displayed the typical apical rhoptry localization but KS-induced parasites (36–40 h.p.i) for all subunits showed an aberrant pattern in most of the induced schizonts. Rather than localizing to the apical tip, the RON12-mCherry marker appeared to localize in internal structures in or around the nascent merozoites, probably in the parasitophorous vacuole (PV) (Figs 5A-C and S5A). To confirm that vesicle transported rhoptry luminal cargo is mistrafficked upon HOPS/CORVET functional depletion, we next undertook immunofluorescence assays with control and VPS16 and VPS18-KS schizonts using antibodies specific against RAP1 (a soluble rhoptry bulb protein) [59]. Consistently, in KS parasites RAP1 did not show the typical rhoptry apical localization observed in controls but instead was found to accumulate around the nascent merozoites likely in the PV (Figs 5D and S5E) similar to the localization of RON12-mCherry. Interestingly, we observed no apparent change in the localization of ARO-mCherry upon KS of all subunits (Fig 5E-5G). In contrast to RAP1 and other luminal rhoptry proteins that are guided by their N-terminal signal peptide and trafficked through the secretory pathway, ARO is recruited to the cytosolic side of the rhoptry bulb membrane via its N-terminal myristoylation and palmitoylation post-translational modifications [60]. These data suggest that rhoptries are still formed in merozoites upon KS of HOPS/CORVET subunits but vesicular trafficking to the rhoptries and delivery of luminal cargo are disrupted.

## Functional depletion of HOPS/CORVET subunits leads to aberrant localization of microneme proteins

To gain insights into the function of HOPS/CORVET in microneme biogenesis and trafficking, we generated parasite lines expressing the mCherry-tagged micronemal proteins EBA-175 (a protein binding the receptor glycophorin A on the erythrocytes surface during invasion) [61,62] and AMA1 (a microneme secreted, parasite plasma membrane localized protein that is part of the moving junction complex during invasion) [63,64] and assessed what impact functional depletion of VPS11, VPS16 and VPS18 had on micronemal protein localization. C2- arrested control and KS (36–40 h.p.i) schizonts were visualized microscopically and the localization of EBA-175 and AMA1 foci was scored. In contrast to the consistent EBA-175 apical localization in controls, most of the KS schizonts showed an aberrant EBA-175 distribution for all targeted core subunits (VPS11-KS: 68.4% ±1.3% (Figs 6A and S5B);

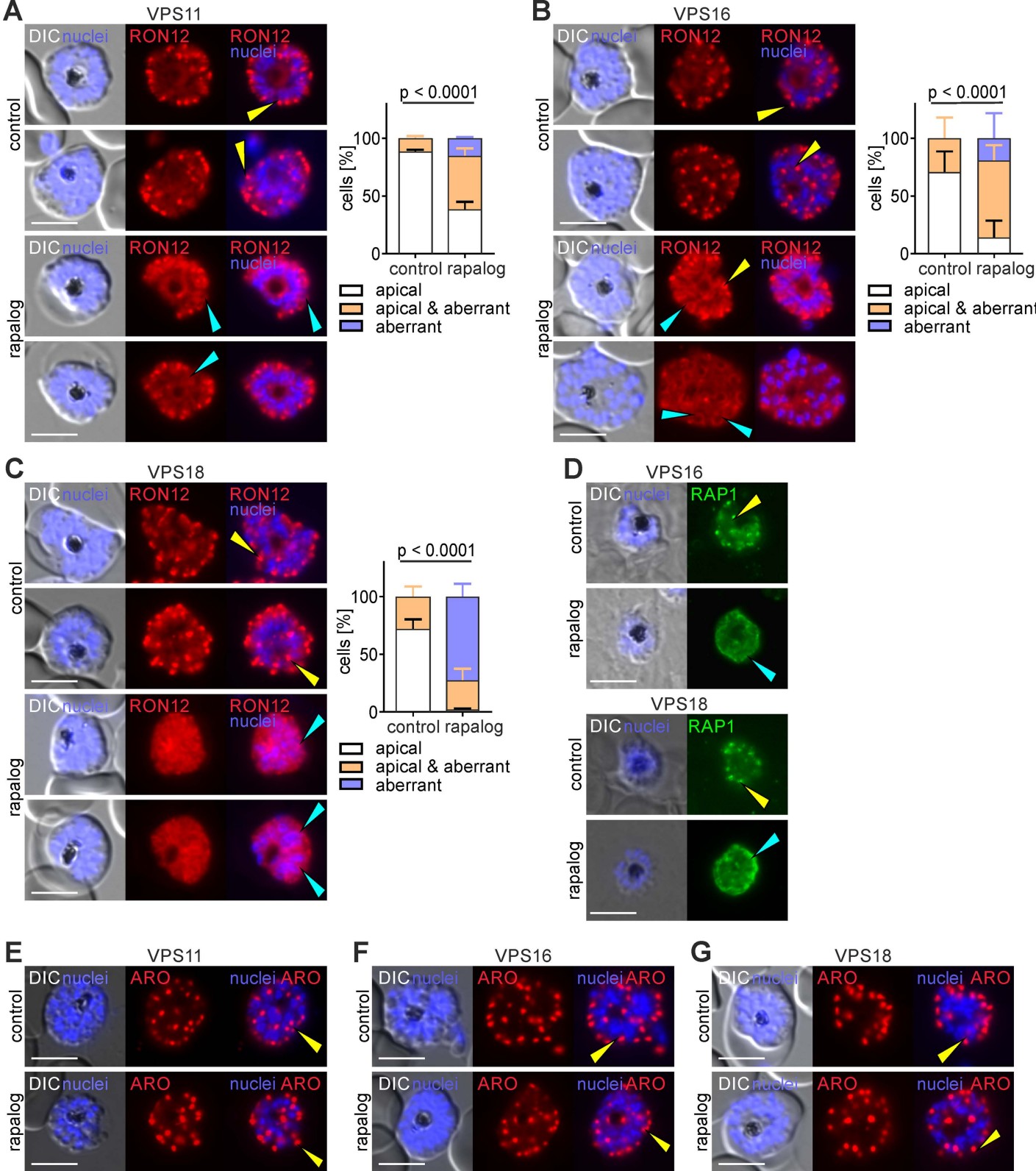

**Fig 5. Functional depletion of HOPS/CORVET subunits causes mislocalization of rhoptry resident proteins. (A, B, C)** Left, live-cell fluorescence microscopy images of C2-arrested control and KS-induced (rapalog) schizonts expressing RON12-mCherry (rhoptry neck luminal protein). Scale bars: 5 μm. Right, quantification of the number of cells with an apical, aberrant and apical & aberrant distribution of RON12. Results from n=3 independent experiments with a total of 75

(control) and 93 (rapalog) VPS11 parasites (A); 99 (control) and 113 (rapalog) VPS16 parasites (B); 126 (control) and 169 (rapalog) VPS18 parasites (C). Coloured error bars indicate SD of every phenotype. p values from a Chi-square test are indicated. **(D)** Immunofluorescence images of VPS16 and VPS18 C2-arrested control and KS-induced schizonts (rapalog) probed with anti-RAP1 (rhoptry luminal bulb protein). Nuclei stained with DAPI. DIC, differential interference contrast. Scale bars: 5 μm. **(E, F, G)** Live-cell fluorescence microscopy images of C2-arrested control and KS-induced (rapalog) schizonts expressing ARO-mCherry (rhoptry bulb cytosolic protein). Nuclei stained with DAPI. DIC, differential interference contrast. Scale bars: 5 μm. Yellow arrows show a typical rhoptry apical localization, light blue arrows show an aberrant localization: diffuse around or in the merozoites, PV or around the nucleus. Additional examples including magnifications are available in S5A.

VPS16-KS: 96.3% ± 4.0% (Figs 6B and S5B); VPS18-KS: 94.8% ±3.6% (Figs 6C and S5B)). In these VPS-KS parasites, EBA-175 was closely associated with the nucleus or was localized diffusely in the merozoites or at the plasma membrane (Figs 6A-C and S5B, S5E). This phenotype was particularly pronounced in VPS18-KS parasites, while EBA-175 localization in VPS11-KS and VPS16-KS schizonts showed often a mixed phenotype between apical foci and the aberrant nuclear/cytosol localization. A similar phenotype was observed for AMA1 with a significant increase in schizonts with aberrantly localized AMA1 upon KS. The localization of AMA1 after VPS11-KS, VPS16-KS and VPS18-KS was diffuse at or near the plasma membrane in the parasite cytosol and not confined to the apical end as expected (Figs 6D-6F and S5C). To distinguish if the observed phenotype was due to cargo trafficking defects or a defect in the biogenesis of the micronemes, we expressed in the VPS16 and VPS18 lines the acylated pleckstrin homology (PH) domain-containing protein (APH)-mCherry, a protein anchored at the surface of the micronemes via N-terminal myristoylation and palmitoylation [65]. VPS16 and VPS18 KS had no apparent effect on the localization of APH-mCherry, as this marker showed the typical apical localization in both control and KS parasites (Figs 6G and S5D). It can be concluded that similar to the phenotype observed in rhoptries, HOPS/CORVET functional depletion causes a defect in trafficking of secreted microneme proteins but has no apparent defect in the biogenesis of the organelles. Combined, these data indicate that KS of HOPS/CORVET subunits at late stages of parasite development leads to the mistrafficking of secreted rhoptry and micronemal proteins away from these invasion organelles. The inability to correctly secrete these key invasion ligands is expected to result in a reduced invasion efficiency and likely explains the observed invasion defect.

## HOPS/CORVET are important for correct rhoptry biogenesis

After showing a mistrafficking of luminal rhoptry proteins when HOPS/CORVET subunits are functionally depleted, but with evidence that rhoptries still form despite disruption of vesicular cargo delivery, we next assessed if there was any evidence of a defect in rhoptry biogenesis upon VPS16-KS and VPS18-KS. We performed expansion microscopy (U-ExM) [56] in combination with NHS-ester staining to visualize rhoptries in VPS16 and VPS18-KS induced parasites. Control schizonts displayed a rhoptry pair with presence of a bulb and a neck associated to an apical ring (Fig 7A and 7B). VPS16-KS and VPS18-KS parasites showed also a rhoptry pair with no discernible structural alterations (Fig 7A and 7B) and the number of rhoptries per nuclei was not altered compared to controls (Fig 7C), however numerous vesicular structures (NHS vesicle-like foci) had accumulated in the KS schizonts (Fig 7A, 7B and 7D) that were in a lower number in the controls (Fig 7D), recapitulating the RON12-mCherry and RAP1 localization seen without expansion (Figs 5A-D and S5A, S5E) and supporting a defect in trafficking to the nascent organelles. Next, we measured the volume of single rhoptries in control and KS-induced schizonts. Individual rhoptries of VPS16 and VPS18-KS parasites exhibited a significantly lower volume than those of controls (Fig 7E and 7F), indicating that the reduction of rhoptry cargo arriving at the organelles after KS of HOPS/CORVET impacts normal development of the rhoptries.

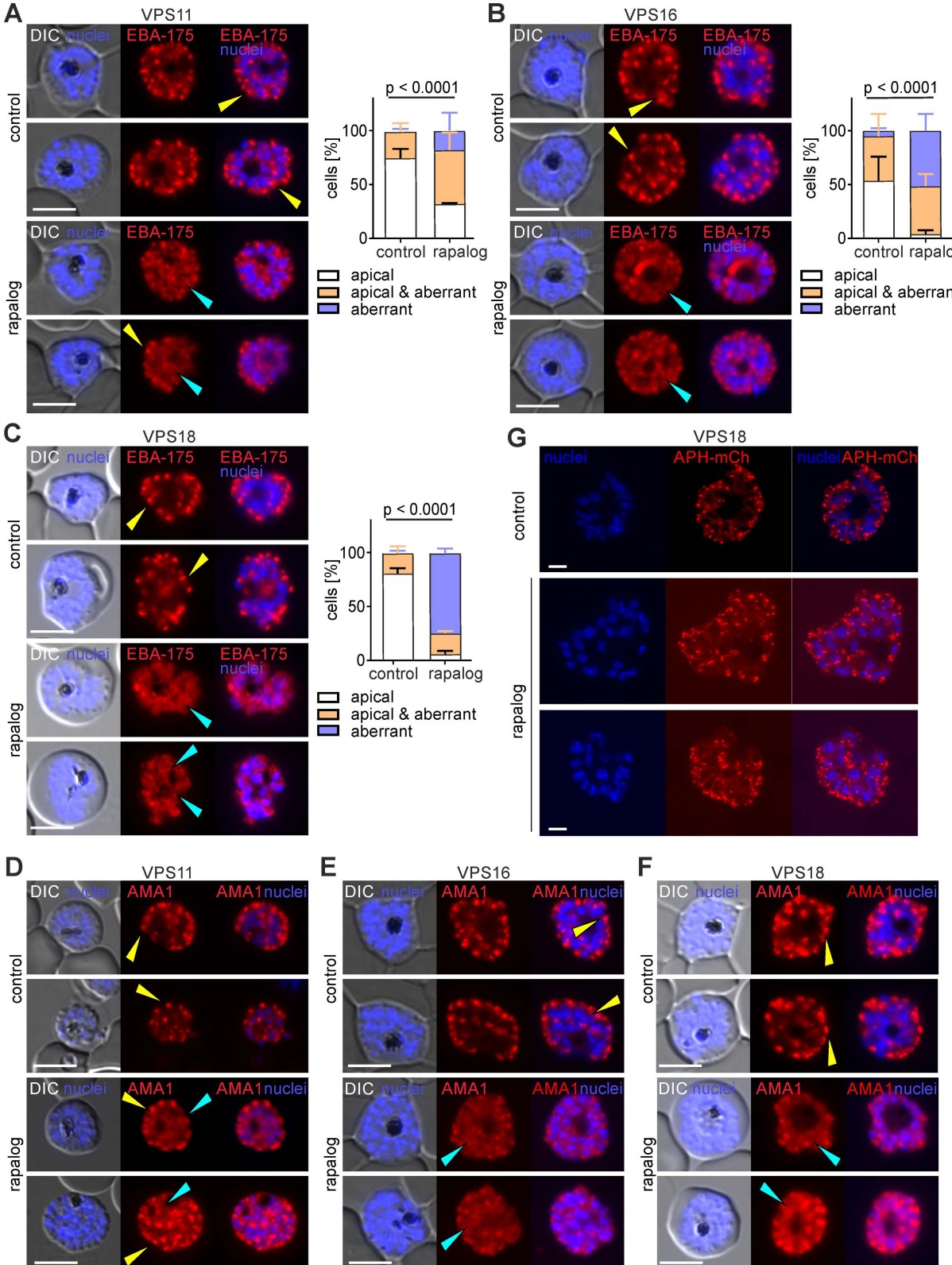

**Fig 6. Functional depletion of HOPS/CORVET subunits leads to aberrant localization of microneme proteins. (A, B, C)** Left, live-cell fluorescence microscopy images of C2-arrested control and KS-induced (rapalog) schizonts expressing EBA175-mCherry (transmembrane microneme protein) Scale bars: 5 μm. Right, quantification of the number of cells with an apical, aberrant and apical & aberrant distribution

of EBA175. Results from n = 3 independent experiments with a total of 150 (control) and 161 (rapalog) VPS11 parasites (A); 97 (control) and 105 (rapalog) VPS16 parasites (B); 112 (control) and 86 (rapalog) VPS18 parasites (C). Coloured error bars indicate SD of every phenotype. p values from a Chi-square test are indicated. **(D, E, F)** Live-cell fluorescence microscopy images of C2-arrested control and KS-induced (rapalog) schizonts of the indicated cell lines expressing AMA1-mCherry (transmembrane microneme protein). Nuclei stained with DAPI. DIC, differential interference contrast. Scale bars: 5 µm. Yellow arrows show a typical microneme apical localization, light blue arrows show an aberrant localization: diffuse around and in the merozoites or around the nucleus. Additional examples including magnifications are available in S5B. **(G)** Confocal live-cell fluorescence microscopy images of C2-arrested control and KS-induced (rapalog) schizonts of VPS18 parasites expressing APH-mCherry (cytosolic surface microneme protein). Nuclei stained with DAPI. Scale bars: 2 µm.

## Discussion

Eukaryotic cells require intracellular membrane fusion for vesicle-mediated uptake and transport of material within and outside the cell. The HOPS/CORVET multisubunit tethering complexes are heterohexameric multitasking hubs with critical functions in vesicular fusion along the eukaryotic endolysosomal and secretory systems [16]. Tethering complexes are localized mostly on the target membrane and sample the environment for the presence of the correct vesicles containing specific Rab GTPases and SNARE proteins. Upon interaction with Rab GTPases on both membranes, these complexes bring the two membranes into closer contact and enhance the SNARE-mediated membrane fusion [66,67]. Each HOPS/CORVET complex typically consists of a core of 4 components (VPS11, VPS16, VPS18 and VPS33), with additional accessory subunits that interact with specific Rab GTPases and SNARE proteins (Fig 1A). Current models suggest that CORVET associates with early endosomes which include those that transport via endocytosis material into a cell, and HOPS associates with late and Golgi-derived endosomes that fuse with the lysosome [68].

In the apicomplexan parasite *T. gondii*, HOPS/CORVET play an essential role in the biogenesis of secretory organelles required for host-cell invasion and parasite gliding motility. Knocking-down of the HOPS/CORVET core subunits VPS11 and VPS18 [35] revealed that these components are required for the targeted transport of rhoptry, microneme and dense granule proteins. Depletion of HOPS accessory subunit VPS39 [35] and CORVET VPS8 [23] also resulted in a defective trafficking to and biogenesis of apical organelles. However, there was no reported evidence that the HOPS/CORVET complex has a role in endocytosis linked to parasite survival. Thus, in *T. gondii* at least, it appears that HOPS/CORVET has been repurposed to enable the *de novo* synthesis of essential secretory organelles and if endocytosis is important for parasite growth, as recently demonstrated [36,37], the HOPS/CORVET complex is not critical for this function. Here we show that in malaria parasites HOPS/CORVET components maintain the canonical endocytosis function that enables hemoglobin vesicular transport and delivery to the digestive vacuole but also have been repurposed for a key role in transporting protein cargo to secretory organelles required for host-cell invasion.

Host cell cytosol (HCC) endocytosis is a key cellular pathway for parasite survival [38] since it provides amino acids from the hemoglobin proteolysis in the digestive vacuole [69] and also free space in the host cell cytosol for parasite growth and to maintain osmotic balance [70]. The parasite specific pathway for HCCU has been partially characterised with several canonical factors [39,53] involved as well as divergent specific parasite proteins [54,71]. However, the role of the HOPS/CORVET tethering complex in this process had not been demonstrated until this study. Here we showed that the core proteins VPS11 and VPS18 are needed for HCC vesicular transport to the DV, as functional depletion of these proteins led to a build-up of intracellular vesicles and failure to deliver hemoglobin into the parasite´s DV. This phenotype is consistent with functions reported in yeast and in mammalian cells where depletion of core subunits caused a defect in late endosome–lysosome fusion [72–74]. In the case of the malaria parasite, the DV could be considered to be a lysosome-like organelle and our results suggest

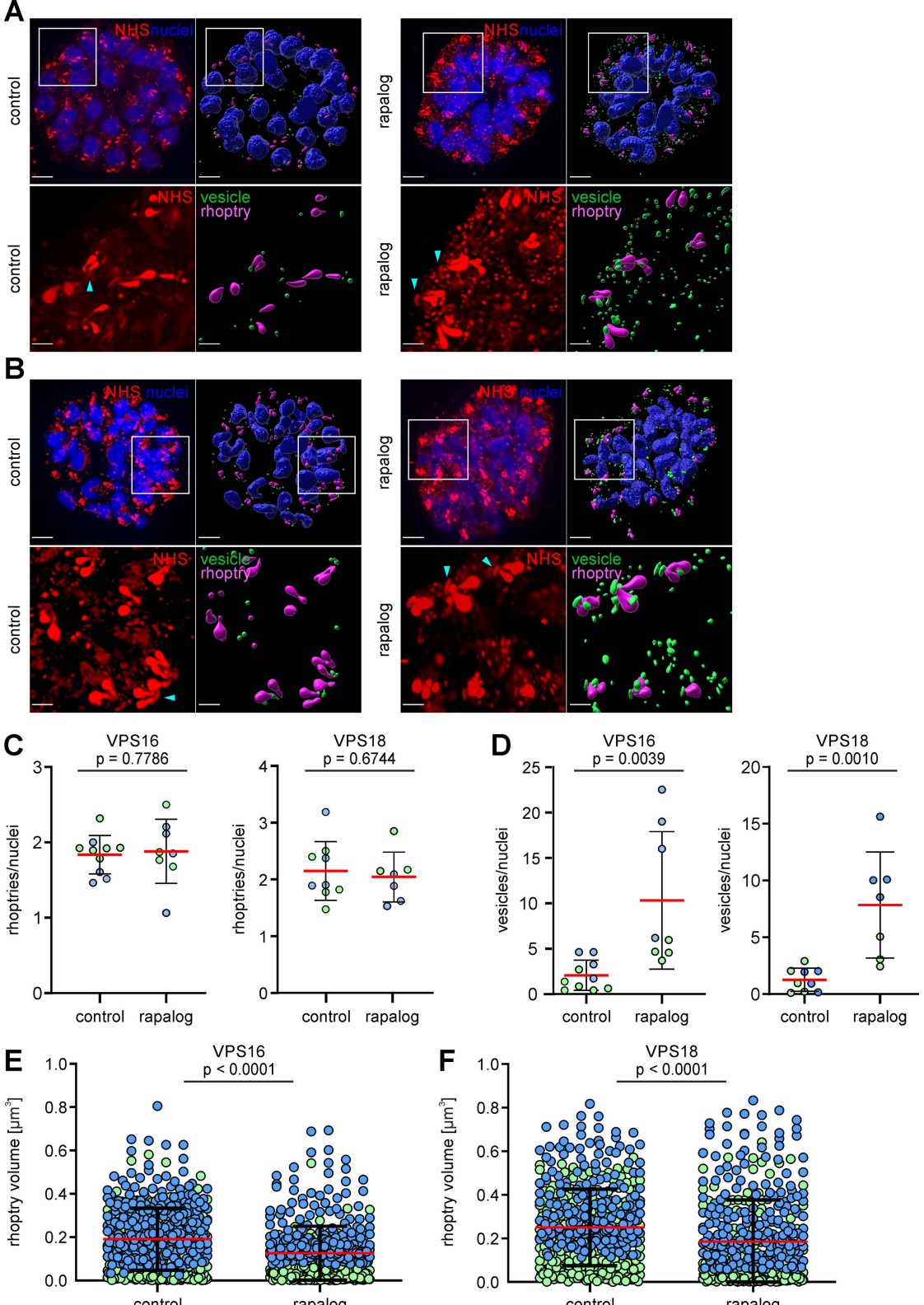

**Fig 7. Functional depletion of HOPS/CORVET subunits causes defect in vesicular trafficking to the rhoptries (A, B) Left panel, confocal images of control and KS-induced VPS16 (A) and VPS18 (B) schizonts prepared with U-ExM and stained with NHS-ester (rhoptries) and DAPI (nuclei).** Right panel, Imaris-generated 3D reconstructions shows nuclei (blue), rhoptries

(magenta) and vesicles (green). Scale bars: 3 μm. Lower panels show enlargement of the region delimited by box in the upper panel. Zoom factor: 300%. Scale bars: 1 μm. Light blue arrows show apical ring **(C)** Number of rhoptries per nuclei in control and KS-induced schizonts (rapalog) analysed by U-ExM. Results from n = 2 independent experiments (distinguished by colours) with a total of 10 (control) and 8 (rapalog) VPS16 parasites; 9 (control) and 7 (rapalog) VPS18 parasites. Mean, red line; error bars, SD. p values from a two-tailed unpaired t-test are indicated. **(D)** Number of vesicles per nuclei in control and KS-induced schizonts (rapalog) analysed by U-ExM. Results from n = 2 independent experiments (distinguished by colours) with a total of 10 (control) and 8 (rapalog) VPS16 parasites; 9 (control) and 7 (rapalog) VPS18 parasites. Mean, red line; error bars, SD. p values from a two-tailed unpaired t-test are indicated. **(E, F)** Rhoptry volume of control and KS-induced schizonts (rapalog) of the indicated cell lines analysed in C and D. Results from n = 2 independent experiments with a total of 493 (control) and 424 (rapalog) rhoptries for VPS16 parasites **(E)**; 480 (control) and 434 (rapalog) rhoptries for VPS18 parasites **(F)**. Mean, red line; error bars, SD. p values from a two-tailed unpaired t-test are indicated.

that VPS11 and 18 play a role in the fusion of HCC containing vesicles with the DV membrane rather than in the initial steps of cytostome formation and HCC uptake and transport. Previous studies have shown that these endocytic vesicles often have a double membrane and are found in close contact with the DV [39,75], suggestive of tethering between the outer vesicle membrane and the DV membrane. Functional depletion of VPS11 and VPS18 phenocopied the hemoglobin trafficking disruption seen with VPS45 [39], Rab5b and Rbns5 [53]. In agreement, we observed co-localization of Rab5b and Rbns5 with VPS11 in the DV membrane and vesicle-like foci. In yeast, CORVET is known to interact with Rab5 proteins in early endosomes and is critical for endosome maturation and fusion to late endosomes [10,18], whereas HOPS interacts with Rab7 proteins in late endosomes and is important for their fusion to lysosomes [76]. Our data suggest that a CORVET-like or HOPS-like complex may play a role in endosomal trafficking in malaria parasites, but the specific subunits, besides the core VPS11 and VPS18, and the interaction with the corresponding Rab proteins, remains to be elucidated.

Since the core HOPS/CORVET subunit protein VPS33 proved refractory to modification, it was not possible to determine whether this protein has an important role in endosomal trafficking or formation of the secretory organelles in malaria parasites. Interestingly, functional depletion of the core subunit VPS16 did not appear to impact endosomal trafficking to the DV. While it is generally supposed that HOPS/CORVET act as a complex, deletions of the individual components have shown different phenotypes in model eukaryotes [21], suggesting that the core components might have additional functions in the cell. In *Drosophila melanogaster* it has been demonstrated that a so-called mini-CORVET consisting of VPS16, VPS18 and VPS33 is involved in endocytosis [43]. While the apparent essential core proteins in the CORVET-like tethering complex of malaria (VPS11, VPS18 and putatively VPS33) are not the same as for the *D. melanogaster* mini-CORVET (VPS16, VPS18, VPS33), the reduced complex size demonstrates that not all core proteins are essential for HOPS/CORVET function in all cases. Additionally, HOPS/CORVET are dynamic tethering complexes known to exchange subunits and form hybrid complexes to perform multiples modes of endocytosis [77].

Our study also demonstrates that all three genetically modified core HOPS/CORVET proteins are essential for correct biogenesis of the apical secretory organelles, the rhoptries and micronemes, of the malaria parasite. For VPS16, disruption of correct trafficking of luminal rhoptry and microneme proteins to their respective organelles, and subsequent inability for the malaria merozoite to invade its next RBC, was the dominant phenotype independent of the time of induction. For VPS11 and VPS18, induction of KS needed to be timed to the last ~10 hrs of intracellular parasite development and when the rhoptry and micronemes begin to be formed. If induced earlier, the dominant phenotype was a failure to grow and form merozoites, suggesting that VPS11/VPS18 dependent endocytosis is essential for parasite development through the first ~36 hrs and its disruption might cause 'traffic jams' that impact segregation and the biogenesis of secretory organelles [38].

Here, we demonstrate that the repurposing of the core HOPS/CORVET subunits for vesicular transport of protein cargo and to produce secretory organelles *de novo* each cycle of development is shared more broadly across apicomplexan parasites. In *T. gondii*, it has been demonstrated that core components of the HOPS/CORVET tethering complex are essential for correct rhoptry and microneme development and function. In *P. falciparum*, late KS of VPS11, VPS16 or VPS18 led to a distinct phenotype where rhoptry luminal proteins (RON12 and RAP1) were found dispersed in vesicle-like structures in the cytoplasm or distributed around the nascent merozoites likely in the parasitophorous vacuole, reminiscent of phenotypes observed in sortilin-depleted parasites where transport from Golgi to the organelles is impaired [28]. Distinctly, ARO, which localises to the cytosolic face of the rhoptries and is likely trafficked independent of any vesicular trafficking pathway [58,60], still localised to this organelle upon VPS11, VPS16 and VPS18-KS. In *T. gondii* VPS11 knock-down caused a mistrafficking of rhoptry and micronemal proteins, which were secreted constitutively into the PV [35]. Using expansion microscopy, we confirmed that a rhoptry-like organelle is formed after loss of VPS16 and VPS18, but the organelles have a decreased volume compared to those of control parasites, likely to due to a reduced cargo delivery. These data suggest that a rhoptry-like structure can still form in the absence of HOPS/CORVET containing Golgi-derived vesicle docking and release of rhoptry luminal cargo, but it fails to reach the size or obtain the luminal proteins required for function. If HOPS/CORVET containing Golgi-derived vesicles form the bulk of rhoptry and microneme targeted vesicles, then this model marks a variation of the current model for rhoptry formation where the rhoptry is thought to form mostly from Golgi-derived vesicles. However, it remains to be determined whether other tethering molecules fulfil functions in rhoptry biogenesis. Our data are in contrast to the VPS11 knock-down phenotype observed in *T. gondii*, where rhoptries were absent or malformed [35]. However, we cannot exclude that at the time of the KS induction in our experiments partial rhoptry formation had occurred [56] and that HOPS/CORVET have a more significant role in the early steps of rhoptry biogenesis.

After KS of VPS11,16 and 18, microneme proteins showed a diffuse localization or were found around the nucleus of nascent merozoites (EBA175) and/or diffusely at the plasma membrane (AMA1) suggestive of defective trafficking to the micronemes. The data using APH, a protein localized to the microneme surface via N-terminal myristoylation and palmitoylation [65], indicate that the micronemes are still formed in HOPS/CORVET deficient parasites but transport of proteins dependent on vesicular trafficking is disrupted. In *T. gondii* targeting of some microneme proteins was severely (MIC3 and MIC8) or partially (AMA1) or not affected (MIC2), suggesting the presence of different microneme populations or distinct trafficking pathways [35].

HOPS/CORVET deficient merozoites were unable to invade new red blood cells, a phenotype that can be explained by the mistrafficking of rhoptry (RONs) and microneme proteins (AMA1, EBAs, Rh5) necessary for interaction with host cell receptors and tight junction formation [9]. Interestingly, egress was not affected in KS parasites, suggesting that exoneme formation and secretion is not dependent on HOPS/CORVET and other tethering molecules might be involved. This was also the case for protein export and trafficking of PVM proteins (EXP2) (S4F and S4G Fig), indicating that HOPS/CORVET do not play a role in vesicular fusion along the classical secretory pathway. In yeast, HOPS has been implicated in transport of proteases from the trans-Golgi to the lysosome via AP-3 coated vesicles [22,78]. Consistently, we co-localized VPS11 with AP-3 in foci-like structures but the role of HOPS subunits in transport of proteases to the DV was not investigated.

In yeast, the molecular architecture of the CORVET and HOPS complex has been described in detail [41,42]. While in *T. gondii* two distinct functional complexes exist [23] *in*

*P. falciparum* the apparent lack of the annotated accessory subunits VPS8 (CORVET) and VPS39/VPS41 (HOPS) that distinguish these two complexes and define the tethering complex selectivity to act as a CORVET-like (early endosome, interacts with Rab5) or HOPS-like (late endosome/secretory organelle, interacts with Rab7 and AP-3) suggests that the composition in *Plasmodium* differs from model organisms. Tethering complexes present in other eukaryotes such as the exocyst [31], DsI1 [34] and CHEVI [16] appear to be absent or reduced in *P. falciparum* parasites [32,33]. It can be speculated that either the complex is reduced to the core components or unknown accessory factors may accomplish the interaction with the corresponding Rab and SNARE proteins. In this study, functional depletion of VPS3 (a CORVET subunit binding Rab5 GTPases [18]) did not result in a profound growth defect and did not recapitulate the phenotype observed in VPS11/16/18 KS parasites. A dispensable function of VPS3 is also in agreement with the piggyBac screen [79] that predicts the gene as likely dispensable. Recently, VPS45, Rbns5 and Rab5b were identified as critical factors for the endolysosomal delivery of HCC to the DV [39,53], suggesting that these proteins may interact with HOPS/CORVET for the tethering of vesicles to the DV membrane. BLAST analysis suggested the presence of a VPS41/ VPS8-like protein (PF3D7_0916400) in the *P. falciparum* genome, a gene predicted to be refractory to disruption in functional screens [79], that might be an accessory protein of the complex. In *T. gondii* a BEACH domain containing protein (BDCP, TGME49_263000) has been identified as part of HOPS/CORVET in association with Rab5 proteins [23]. The homologue (PF3D7_1124100) is annotated in the *P. falciparum* genome and is predicted to be indispensable for parasite development [79]. Further functional analyses on these proteins are necessary to reveal the role in endocytosis and organelle biogenesis as well as proteomic analyses that define the composition of the complexes at different time points of development.

Our data support that HOPS/CORVET have a critical stage-specific dual function in the intracellular development of *P. falciparum* parasites. We confirmed that conserved endolysosomal factors are utilized in malaria parasites for canonical functions (e.g., endocytosis and trafficking of host cell material) but also to accomplish the parasite specific function of protein cargo trafficking to apical secretory organelles, necessary for invasion and proliferation in new red blood cells. This reorganization of the HOPS/CORVET complexes provides insights into novel dual functions of a canonical eukaryotic cellular pathway that have evolved to fulfil the needs of the malaria parasite complex life cycle.

## Materials and methods

### Cloning of DNA constructs

To modify the endogenous locus of the core HOPS/CORVET subunits SLI plasmids were generated as previously described [45]. The last 1000 bp of the genes *vps3* (PF3D7_1423800), *vps11* (PF3D7_0502000), *vps16* (PF3D7_1239900) and *vps18* (PF3D7_1309700) encoding for the C-terminus of the protein were amplified from 3D7 genomic DNA using primers listed in S1 Table. Homology regions were cloned using T4 ligation or Gibson assembly [80] into a modified version of pSLI-2xFKBP-GFP-2xFKBP-Sandwich plasmid using AvrII and NotI. This plasmid expresses the mislocaliser 1xNLS-FRB in tandem fused to the human dihydrofolate reductase via a skip peptide [53] from a second cassette under the *nmd3* promoter.

For co-localisation experiments the following plasmids were transfected in the integrant cell lines: p$^{crt}$-P40PX-mCherry-BSD [39], p$^{nmd3}$-mCherry-K13-DHODH [71], p$^{crt}$-pGRASPmCherry-BSD [39], p$^{sfa32}$-Rabenosyn-5-mCherry_DHODH [53], p$^{sfa32}$-Rab5b-mCherry-yDHODH [53] and p$^{hsp86}$mScarlet-Rab7-yDHODH [81] (kindly provided by Dave Richard). With the aim to co-localize the AP-3μ subunit and mScarlet-Rab5a in the

VPS11-2xFKBP-GFP-2xFKBP line the full-length genes (*ap-3μ*, PF3D7_1440700 and *rab5a*, PF3D7_0211200) were amplified from 3D7 gDNA using the primers listed in S1 Table and cloned into crt-P40-mScarlet_nmd3-NLS-FRB-T2A-DHODH [53] using Gibson cloning assembly and XhoI and AvrII restriction enzymes for AP-3μ (C-terminal tagging) and KpnI and XmaI for Rab5a (N-terminal tagging).

To analyse the content of vesicles in KS -induced parasites, a plasmid was generated to express the first 108 amino acids of glycophorin binding protein 130 (GBP130, PF3D7_1016300) under the *crt* promoter after exchanging the *nmd3* promoter region in p^nmd3 SBP1mScarlet [82] with *crt* using NotI and XhoI. Next, the sequence coding for GBP (1–108 aa) was amplified from 3D7 gDNA using primers listed in S2 Table and cloned into the p^crt-SBP1mScarlet using XhoI and AvrII.

To generate the plasmids for co-localisation experiments while simultaneously carrying out knock sideways, *ama1*AMA1mCherry, *ama1*AROmCherry and *ama1*IMC1 cmCherry were amplified from p^ama1AMA1mCherry, p^ama1AROmCherry or p^ama1IMC1 cmCherry [83] by PCR and cloned into ama1-p40-mSca_nmd3'-NLS-FRB-T2A-yDHODH [54] using the KpnI/XmaI restriction sites, resulting in *ama1*AROmCherry_*nmd3*'-NLS-FRB-T2A-yDHODH, p*ama1*IMC1 cmCherry_*nmd3*'-NLS-FRB-T2A-yDHODH and *ama1*AMA1mCherry_*nmd3*'-NLS-FRB-T2A-yDHODH. To visualize RON12 upon HOPS/CORVET KS p*ama1* RON12mCherry [83] was transfected in the corresponding cell lines. To express EBA175 under the *ama1* promoter the full-length EBA175 gene was synthesized (Gene Script) with KpnI and AvrII restriction sites, digested and cloned into p^ama1AMA1mCherry-yDHODH, resulting in p^ama1EBA175-mCherry-yDHODH. p^ama1APH-mCherry was generated by amplifying the full-length sequence of APH (PF3D7_ 0414600) from 3D7 cDNA by using primers listed in S2 Table and cloned into p^ama1-AMA1 mCherry [83] using AvrII and KpnI.

## Parasite culture and synchronisation

Asexual blood stages of *P. falciparum* (strain 3D7) parasites were cultured in RPMI medium (Gibco) containing 0.5% Albumax (Gibco), 200 mM hypoxanthine and 2–5% fresh human RBCs (B+; provided by Universität Klinikum Eppendorf, Hamburg). Cultures were maintained at 37 °C, in a microaerophilic atmosphere of 1% $O_2$, 5% $CO_2$ and 94% $N_2$ following standard procedures [84].

To obtain highly synchronous *P. falciparum* cultures, late schizont stages were isolated in a percoll gradient [85] using a 60% percoll solution, washed once with RPMI and incubated with fresh erythrocytes shaking at 37 °C for 30 minutes. Schizont stages were further cultured for 4 or 8 hours under standard conditions, followed by synchronisation with 5% sorbitol [86] to eliminate late stages and obtain ring stage parasites with a 0–4 or 0–8 hours post invasion (h.p.i) age window. For growth curves, asynchronous cultures were synchronised with 5% sorbitol and adjusted to the desired ring stage parasitemia.

## Parasite transfection

For transfection of episomal constructs, mature schizonts were enriched using a 60% Percoll density gradient and electroporated in transfection buffer containing 50 μg of purified plasmid DNA using an Amaxa Nucleofector II (Lonza, AAD-1001N, program U-033) [87]. Selection of transgenic parasites was performed either with 4 nM WR99210 (Jacobus Pharmaceuticals, USA), 2–5 μg/ml Blasticidin S (Life Technologies), or 0.9 μM DSM1 (BEI Resources, https://www.beiresources.org/) depending on the resistance marker. For generation of stable integrant cell lines via SLI, an asynchronous culture of parasites (5–10% parasitemia) containing the episomal SLI plasmids was cultured with 400 μg/ml G418 (Sigma-Aldrich) to select for

integrants carrying the desired genomic modification as described previously [45]. To confirm correct genomic integration of the SLI plasmid, genomic DNA from parasites selected under G418 was prepared with a QIAamp DNA Mini Kit and analyzed by PCR using primers specific for the 5' and 3' integration junctions and primers to detect the original locus (See S2 Table for integration check primers).

### Stage specific knock sideways (KS) induction

To induce mislocalization of the HOPS/CORVET subunits at specific time points mature schizont stages were purified using percoll 60% as described above and allowed to invade for 30 minutes shaking at 37 °C in RPMI medium containing fresh RBCs. Sorbitol-synchronised 0–4 h.p.i ring stage cultures were resuspended in RPMI medium and split into two dishes, one served as control, without rapalog and to the other, rapalog (Clontech) was added to a final concentration of 250 nM at 0–4 h.p.i for early knock-sideways induction or at 32–36 h.p.i for late KS induction. For VPS18-2xFKBP-GFP-2xFKBP KS was induced at 36–40 h.p.i to avoid defects on schizont segmentation.

### Growth curves and flow cytometry

Flow cytometry was used to quantify parasitemia and assess growth in parasite cultures as previously described [83,88] with some modifications. Ring stage parasite cultures were sorbitol synchronised as described above and parasitemia was adjusted to 0.1–1% and split into three 2 mL dishes, one treated with 250 nM rapalog (Clontech) at ring stage, one at 32–36 h.p.i and one left as control. Culture media was changed daily (fresh rapalog added to the KS-induced culture) and parasitemia was determined every day for four days (two replication cycles) by flow cytometry. For determination of ring stage parasitemia, 20 μL of resuspended culture was incubated in 80 μL RPMI containing 25 μg/mL SYBR Green (Sigma-Aldrich) for 40 min in the dark at 37°C. Samples were washed once and incubated for another 40 min at 37°C in RPMI. The final sample was resuspended in 120 μL RPMI and measured with an ACEA NovoCyte flow cytometer. For late stage parasitemia, 20 μL of resuspended culture was incubated with 80 μL RPMI containing 4.5 μg/ml dihydroethidium (DHE) (Cayman Chemical) and 5 μg/mL SYBR Green (Sigma-Aldrich) for 20 min at 37°C and measured with an ACEA NovoCyte flow cytometer by counting 100 000 events. A corresponding Giemsa-stained thin blood smear was made for every sample to verify parasitemia by light microscopy.

### Stage quantification

Highly synchronous parasite cultures were synchronised to a 4 h time window as described above. At 0–4 h.p.i cultures were split into three 2 mL dishes: one was treated with 250 nM rapalog at 0–4 h.p.i (ring stage), one at 24–28 hpi (trophozoite stage) and one left as control. Giemsa-stained thin blood smears were obtained at 0–8 h.p.i, 16–20 h.p.i, 24–28 h.p.i, 32–36 h.p.i, 40–44 h.p.i, 48–52 h.p.i). For every time point random images of the Giemsa-stained smears were acquired and erythrocytes were counted by the automated 'Parasitemia' software (https://www.gburri.org/parasitemia/ https://www.gburri.org/parasitemia/); 800–3000 erythrocytes were counted per sample. Parasitemia was determined from the acquired images and parasite stages were scored as rings, late rings, early trophozoites, trophozoites, schizonts, segmenters and aberrant (parasites with vesicles or abnormal morphology).

### Live cell microscopy and staining

Live-cell microscopy of parasite cultures was performed as previously described [89]. Briefly, parasite cultures were incubated for 20 min at 37 °C in RPMI containing 1 μg/mL 4',6'-diamidine-2'-phenylindole dihydrochloride (DAPI) (Roche) or 2.5 μM

Bodipy-TR-C5-ceramide (Invitrogen). DIC and fluorescence images of infected erythrocytes were acquired using a Leica D6B fluorescence microscope equipped with a Leica DFC9000 GT camera and a Leica Plan Apochromat 100×/1.4 oil objective or a Zeiss AxioImager M1 or M2 microscope equipped with a LQ-HXP 120 light source and a Hamamatsu Orca C4742-95 camera.

## Confocal spinning disk microscopy

Confocal microscopy was performed using a Nikon TI2 microscope equipped with a Yokogawa CSU-W1 spinning disc and Super Resolution by Optical Pixel Reassignment (SoRa) unit (Nikon). To image the U-ExM samples a ×60 water objective (NA 1.27) was used and images were acquired using a 405 nm (DAPI) and 561 nm (NHS-ester 594) laser. 30–70 Z-stacks were acquired using a 0.15 μm step size. Images obtained were first deconvoluted using the 3D deconvolution function and then denoised with the AI-based denoising function provided in the NIS Elements software (v5.42.03, Nikon) and exported as.nd2 files for further analysis.

## Cell size and nuclei number determination

For cell size measurement, DIC images of synchronous control and KS-induced parasites with an 8 h time window were acquired at 34–42 h.p.i as described above and analysed by Image J (Version 1.48) [90] to measure parasite area (μm²). Analysis of the images was performed blinded to the condition of the sample.

For nuclei number counting, late stage (32–36 h.p.i) control and KS parasites after early (0–4 h.p.i) and late (28–32 h.p.i) KS induction were incubated with 1 μM Compound 2 [55] (kindly provided by Mike Blackman) to prevent egress for 8 h and stained with DAPI for 15 min at 37 °C. Fluorescence images were acquired as described above. Assessment of cell size and nuclei number was undertaken blinded to the sample conditions and the number of nuclei was counted from the DAPI signal.

## Bloated digestive vacuole assay

A bloated vacuole assay was performed following previously described protocols with some modifications [39]. Briefly, asynchronous parasite cultures were sorbitol synchronised to a stage window of 10–18 h.p.i by successive 5% sorbitol treatment 10 h apart. At 18–26 h.p.i cultures were split into two dishes and E64 protease inhibitor (Sigma Aldrich) was added to each dish to a final concentration of 33 μM. Rapalog (250 nM) was added to one dish for KS induction and the other left as control. After 8 h incubation, samples were collected and stained with DHE (4.5 μg/ml) and DAPI (1 μg/μL) in RPMI for imaging. DIC and DHE images were used for scoring parasites and assessment of bloated and non-bloated FV was undertaken blinded to the sample conditions. Parasites with a FV filling more than 1/3 of the cytosol were scored as bloated. At least 25 cells were scored per condition.

## Vesicle accumulation and hemozoin size assay

The accumulation of vesicles and hemozoin size determination was modified from previously published protocols [39]. Asynchronous parasite cultures were sorbitol synchronised twice with a 10h interval to obtain a stage window of 10–18 h.p.i. Cultures were divided into two dishes, one served as control and to the other rapalog was added to a final concentration of 250 nM for KS induction and incubated under standard conditions. Samples were collected 8 h (18–26 h.p.i) and 16 h (26–34 h.p.i) post rapalog addition and imaged immediately to acquire DIC images. The number of vesicles accumulating in the parasite cytosol was counted while

blinded to the sample identity. DIC images were used to measure the area of the hemozoin using Image J [90]. The same protocol was performed for parasites expressing P40XP-mCherry and GBP-mScarlet to visualize overlapping of the reporter with the accumulating vesicles.

## Invasion and egress assay

To determine invasion and egress rate in control and KS-induced parasites previously published protocols were adapted [54,91]. Parasite cultures were synchronised to a defined time window by 5% sorbitol treatment 4 h after controlled invasion of Percoll-purified schizonts. At 32–36 hpi each culture was split into two dishes, one treated with 250 nM rapalog and the other left as control. After 6 h incubation, Compound 2 was added to a final concentration of 1 µM for another 6 h. Arrested schizonts (44–50 h.p.i) were isolated by percoll, washed and allowed to invade fresh red blood cells for 4 h. A sample (post-percoll) was taken and treated with 50 µM E64 (Sigma Aldrich) to determine baseline schizont parasitemia by flow cytometry and Giemsa smears. After invasion was allowed to proceed for 4 h, Giemsa smears and samples for flow cytometry were collected (pre-sorbitol) for ring and schizont parasitemia determination, followed by synchronisation with 5% sorbitol. Ring stage parasitemia was determined by flow cytometry. Invasion (number of rings per ruptured schizont) and egress rate, was calculated as following:

$$\text{rings per ruptured schizont} = \frac{\text{ring parasitemia (post sorbitol)}}{\text{schizont parasitemia (post Percoll)} - \text{schizont parasitemia (pre sorbitol)}}$$

$$\text{egress rate} = 1 - \frac{\text{schizont parasitemia (pre sorbitol)}}{\text{schizont parasitemia (post percoll)}} x\ 100$$

## Rhoptry and microneme protein localization

To assess localization of microneme (AMA1-mCherry, APH-mCherry and EBA175-mCherry) and rhoptry (ARO-mCherry and RON12-mCherry) proteins in control and KS -induced parasite lines expressing these organelle markers, parasite cultures were tightly synchronised as described above to obtain a 4h-time window. At 32–36 h.p.i cultures were divided into two dishes; KS was induced in one dish by addition of rapalog and one was left as control. C2 was added 4 h later to a final concentration of 1 µM to prevent egress and the cultures incubated for a further 6 h. Samples were collected, stained with DAPI and immediately imaged. Fluorescence microscopy images were used to score for localization of the rhoptry and microneme proteins and localization classified as apical, aberrant (diffuse in parasite cytosol, parasite periphery, around the nucleus) and mixed (apical and aberrant). Scoring was performed while blinded to the sample identity. At least 20 parasites per condition were imaged from three independent replicates.

## Immunofluorescence assays

C2-arrested control and KS-induced schizonts prepared as described above were harvested, washed in PBS, and fixed with 4% paraformaldehyde/0.0075% glutaraldehyde in PBS [92]. Cells were permeabilized with 0.1% Triton X-100 in PBS, blocked with 3% BSA in PBS, and incubated overnight at 4 °C with primary antibodies: mouse anti-RAP1[93] (1:500) and rat anti-RFP (Chromotek) (1:500) diluted in 3% BSA in PBS. Cells were washed 3 times with PBS

and incubated for 1 h with Alexa 488 nm and Alexa-594 conjugated secondary antibodies specific for mouse and rat IgG (Invitrogen) diluted 1:2000 in 3% BSA in PBS and containing 1 μg/ml DAPI. Cells were directly imaged after washing 5 times with PBS.

For protein export analysis, schizonts were isolated from synchronous cultures with a percoll gradient. After 30 min incubation in fresh RBCs at 800 rpm and 37 °C, cultures were split into two dishes, one with 250 nM rapalog and one left as control. After 6–8 h.p.i cells were synchronised with 5% sorbitol treatment and samples were collected at the trophozoite stage the day after (22–30 h.p.i). Samples were fixed in 100% acetone and processed as previously published [82]. Samples were labelled with the primary antibodies: mouse anti-EXP2 (1:2500; European Malaria Reagent Repository), rabbit anti-REX1 (1:10000) [82] rabbit anti-SBP1 (1:2500) [82] and the corresponding secondary antibodies: Alexa Fluor 488 nm conjugated goat antibodies against mouse IgG (1:2000; Invitrogen) and Alexa Fluor 594 nm conjugated goat antibodies against rabbit IgG (1:2000; Invitrogen).

## Ultrastructure expansion microscopy

Ultrastructure expansion microscopy (U-ExM) was adapted from recent published protocols [56]. Round coverslips (12 mm, Fisher Cat. No. NC1129240) were treated with poly-D-lysine (1 mg/ml) for 1 h at 37 °C, washed twice with MilliQ water, and placed in the wells of a 12-well plate. Parasite cultures containing 2–5% C2-arrested control and rapalog-induced schizonts were resuspended in 1 ml incomplete RPMI (without Albumax) and allowed to settle down in the coverslip for 30 min at 37 °C. Culture supernatant and excess cells were removed with a pipette before fixation with 500 μL of 4% v/v paraformaldehyde (PFA) in 1xPBS for 20 min at 37 °C. Coverslips were washed carefully three times with warm PBS followed by incubation overnight at 37 °C with 1 mL of freshly prepared 0.7% v/v formaldehyde/1% v/v acrylamide (FA/AA) in PBS.

Monomer solution (19% w/w sodium acrylate (Sigma Cat. No. 408220), 10% v/v acrylamide (Sigma Cat. No. A4058, St. Louis, MO, USA), 0.1% v/v N,N'-methylenebisacrylamide (Sigma Cat. No. M1533) in PBS) was prepared beforehand before gelation, aliquoted and stored at −20 °C until used. Prior to gelation, coverslips were removed from the FA/AA solution, washed once in PBS and allowed to dry briefly. For gelation, 35 μL of monomer solution was pipetted onto a parafilm followed by 5 μL of 10% v/v tetraethylenediamine (TEMED; ThermoFisher Cat. No. 17919) and 5 μL of 10% w/v ammonium persulfate (APS; ThermoFisher Cat. No. 17874). Rapidly, each drop was covered with the coverslip with the cell side facing down to the drop and incubated 5 min on ice for gel penetration. Gels bound to coverslips were incubated at 37 °C for 1 h in a humid chamber and transferred carefully to wells of a 6-well plate containing denaturation buffer (200 mM sodium dodecyl sulphate (SDS), 200 mM NaCl, 50 mM Tris, pH 9). Gels were incubated in denaturation buffer with shaking for 15 min at room temperature to detach them from the coverslip and parafilm. Afterwards, gels were transferred to 1.5 mL tubes containing denaturation buffer and incubated at 95 °C for 90 min. Following denaturation, gels were transferred to Petri dishes containing 10 mL of MilliQ water for the first round of expansion for 30 min and then incubated in fresh MilliQ water overnight at room temperature. Large gels werecut in pieces, frozen and stored at -20 °C. The freezing and thawing procedure is described in detail in [56].

A piece of gel was subsequently shrunk with two rounds of 15 min washes in 1x PBS. For NHS ester-594 staining, gels were incubated for 2.5 h in 500 μl PBS containing 10 μg/ml NHS ester 594 (ThermoFisher Cat. No. A37572) protected from light at room temperature. Gels were washed three times with PBS, followed by three 30 min washes in MilliQ water for a second round of expansion. DAPI was added to the water at a final concentration of 5 μg/ml for nuclei staining. Gels were either imaged immediately following re-expansion or stored in MilliQ water until imaging.

For imaging, a 35 mm imaging dish with a polymer coverslip (Ibidi, Cat No. 81156) were treated with poly-D-lysine (1mg/ml) for 1 h at 37 °C, then washed three times with MilliQ water. Subsequently, stained gels were placed in the coated imaging disk and taken to the microscope for imaging.

## Data analysis and statistics

Statistical analysis was performed with GraphPad Prism 6.03 and details regarding the number of replicates and number of cells analysed per replicate are provided in the figure legends. p values ≤ 0.05 were considered significant. Images were processed with Corel Photo Paint software (version X6) or Fiji [94] to crop the images, to adjust brightness and intensity and to overlay channels. Figures were arranged using CoreIDRAW X6 and Photoshop. Fiji (ImageJ 1.53q) was used for image processing measuring cell and hemozoin size.

For analysis of rhoptries from NHS-ester-stained U-ExM samples a surface identification pipeline on Imaris X64 (v9.8.2, BitPlane AG, Zurich, Switzerland) was established following previous protocols [95]. Denoised and deconvoluted images exported as.nd2 files were converted into.ims format and were analyzed in batch mode to ensure identical processing and quantification. The NHS-ester 594 nm signal, corresponding to rhoptries and vesicles (structures with a high protein content), and DAPI 405 nm signal (nuclei) were segmented and thresholded by intensity to remove background signal. Touching objects were split using a seed point diameter of 0.400 μm and then surfaces were further filtered using the quality parameter provided by the software. To separate the segmented surfaces into rhoptry and vesicles, objects were classified based on their volume, with the vesicles having a much smaller volume than the rhoptries. Objects with a volume lower than 1.3 $\mu m^3$ were considered for analysis. Thresholds were saved and applied across replicates and conditions to keep analysis identical for all images. Once the surfaces were identified and categorized into rhoptry or vesicles, the volume in $\mu m^3$ was quantified for single rhoptries. The total counts of objects and their volume were exported as.csv files. Number of rhoptries and vesicles per nuclei was quantified from individual schizonts. Volume measurements of single rhoptries were analyzed and plotted using GraphPad Prism 6.03.

## Supporting information

**S1 Fig. The HOPS/CORVET core subunits are important for intracellular replication. (A)** Detailed schematic representation of the SLI strategy to modify the genomic locus of the core subunits of HOPS/CORVET and to express simultaneously a nuclear mislocalizer for knock sideways. Binding sites of the oligonucleotides chosen for validation of genomic integration are shown (P1-4). FKBP: FK506 binding protein, dimerization domain; 2A: skip peptide; Neo-R: neomycin phosphotransferase; hDHFR: human dihydrofolate reductase; NLS: nuclear localization signal; FRB: FKBP-rapamycin-binding domain. **(B)** Confirmatory PCR on genomic DNA from 3D7 (wt) and the indicated cell lines to validate correct integration of the SLI plasmid at the 3'- and 5'-end and disruption of the original locus (ori) using combinations of the primers (P1-4) indicated in A. Size in kbp is shown. **(C)** Live-cell fluorescence microscopy images of transgenic parasites expressing the endogenously 2xFKBP-GFP-2xFKBP tagged VPS3 in trophozoites (upper panel) and late stages (lower panel). Nuclei were stained with DAPI. DIC, differential interference contrast. Scale bars: 5 μm. **(D)** Live cell fluorescence microscopy images of transgenic parasite lines expressing endogenously tagged VPS18-2xFKBP-GFP together with GRASP-mCherry as marker for the Golgi apparatus. Nuclei were stained with DAPI. DIC, differential interference contrast. Scale bars: 5 μm. Zoom (300x) of the indicated white boxes is shown to visualise localization of the VPS18 vesicle-like foci in relation to Golgi. **(E)** Live cell

fluorescence images of VPS18-2xFKBP-GFP-2xFKBP parasites co-expressing mScarlet- Rab7 (late endosomes, ELC) and mScarlet-Rab6 (trans-Golgi) and VPS11-2xFKBP-GFP-2xFKBP parasites co-expressing AP3-mScarlet. Scale bars: 5 μm (F) Growth rate (fold increase in parasitemia versus start parasitemia) over two replication cycles of the indicated cell lines in presence (+, red) or absence (−, blue) of rapalog (rapa) calculated from the curves in Fig 1D. Mean (green line) of n = 4 for VPS11, 16 and 18 and n = 3 for VPS3. Error bars (black lines) indicate SD and p values from a two-tailed unpaired t-test are indicated.
(PDF)

**S2 Fig. Efficiency of the knock-sideways system for HOPS/CORVET subunits. (A)** Live fluorescence images of control and KS-induced (rapalog) VPS11-2xFKBP-GFP-2xFKBP parasites co-expressing 1xNLS-FRB-mCherry **(B-E)** Upper panel, quantification of the number of cells showing a digestive vacuole (DV)/ vesicular foci (wild type), nuclear (mislocalized) and nuclear/vesicular foci (partially mislocalized) localization of the indicated VPS subunits in control and KS-induced (rapa) parasites. Results from n = 2 independent replicates with a total of 136 (control) and 123 (rapalog) VPS11 parasites; 136 (control) and 126 (rapalog) VPS16 parasites; 88 (control) and 88 (rapalog) VPS18 parasites. p values from a Chi-square test are indicated. Lower panel, representative fluorescence live cell images of control and KS-induced (rapalog) parasites.
(PDF)

**S3 Fig. Effect of HOPS/COVET subunits knock-sideways on intracellular development. (A, B, C)** Stage distribution of control (−) and KS-induced parasites (+ rapa). Left, stage distribution relative to the total of counted infected red blood cells at the indicated time points (h.p.i). Right, stage distribution relative to the total parasitemia at the indicated time points. Mean of n = 3 independent replicates with at least 50 parasites per time point. Error bars indicate SD. **(D)** Flow cytometry-based determination of parasitemia of synchronised cultures of the indicated parasite lines after one replication cycle without rapalog (control) or when rapalog was added to the rings (0–4 h.p.i, red line) or to late stages (32–36 h.p.i, purple).
(PDF)

**S4 Fig. Effect of HOPS/CORVET subunits knock-sideways on HCCU and protein export. (A)** Images of Giemsa smears of VPS11, 16 and 18 control and KS-induced parasites (rapalog) when rapalog was added to the ring stages. Scale bars: 3 μm. **(B)** Images of DHE (upper panel) and BODIPY-stained (lower panel) control and KS-induced (rapalog) VPS11 parasites. Blue arrows show vesicles. Scale bars: 5 μm. **(C)** Hemozoin size (μm²) of control (- rapa) and KS-induced (+rapa) VPS11-2xFKBP-GFP-2xFKBP parasites at the indicated time points. Mean (green line) of n = 3 independent replicates with a total of 77 control and 66 KS-induced parasites at 18–26 h.p.i and 73 control and 65 KS-induced parasites at 26–34 h.p.i. Error bars indicate SD and p values from a two-tailed unpaired t-test are indicated. **(D)** Enlarged micrographs of control and KS-induced (rapalog) VPS18-2xFKBP-GFP-2xFKBP parasites expressing P40PX-mCherry. Blue arrows show vesicles surrounded by P40PX. Zoom factor 65x **(E)** Live cell images of saponin-lysed control and KS-induced (rapalog) VPS18-2xFKBP-GFP-2xFKBP parasites expressing GBP[1-108]-mScarlet. Blue arrows show overlapping of vesicles and mScarlet signal. Zoom factor 600 x **(F)** IFA images of control and KS-induced (rapalog) VPS18-trophozoites probed with α-EXP2 (parasitophorous vacuole membrane resident protein) and α-SBP1 (Maurer´s clefts resident exported protein). Nuclei stained with DAPI. DIC, differential interference contrast. Scale bars: 5 μm. **(G)** IFA images of control and KS-induced (rapalog) trophozoites of the indicated cell lines probed with α-REX1 (Maurer´s clefts resident exported protein).
(PDF)

**S5 Fig. Knock-sideways of HOPS/CORVET leads to mistrafficking of rhoptry and microneme proteins. (A, B)** Representative live-cell fluorescence microscopy images of C2-arrested control and KS-induced (rapalog) VPS11 (left) VPS 16 (middle panel) and VPS18 (right) schizonts expressing RON12-mCherry (rhoptry neck luminal protein) **(A)** and EBA175-mCherry (transmembrane microneme protein) **(B)**. Scale bars: 5 µm. Enlarged micrographs (zoom: 600x) of the indicated white boxes are shown right of each panel to visualise localization of organelle markers. Yellow arrows show a typical microneme or rhoptry apical localization, light blue arrows show an aberrant localization: diffuse around and in the merozoites, PV (RON12) or around the nucleus. Nuclei stained with DAPI. DIC, differential interference contrast. Scale bars: 5 µm. **(C)** Confocal live-cell fluorescence microscopy images of C2-arrested control and KS-induced (rapalog) schizonts of VPS11 parasites expressing AMA1-mCherry. Nuclei stained with DAPI. Scale bars: 2 µm. **(D)** Live-cell fluorescence microscopy images of C2-arrested control and KS-induced (rapalog) schizonts of VPS16 parasites expressing APH-mCherry (cytosolic surface microneme protein). Nuclei stained with DAPI. Scale bars: 5 µm. **(E)** Immunofluorescence images of C2-arrested control and KS-induced (rapalog) VPS16 schizonts expressing EBA175-mCherry probed with anti-RAP1 (rhoptry luminal bulb protein) and anti-mCherry (RFP) (EBA175, microneme transmembrane protein). Nuclei stained with DAPI. DIC, differential interference contrast. Scale bars: 5 µm. Enlarged micrographs (zoom: 600x) of the indicated white boxes are shown right to visualise apical co-localization of both organelle markers (yellow arrows) and PV (light blue arrows) of RAP1 in contrast to the internal staining of EBA175 mCherry in KS-induced schizonts. (PDF)

**S1 Table. Comparative bioinformatics analysis of genes coding components of CORVET and HOPS complexes in *Plasmodium falciparum and Toxoplasma gondii*.** *Saccharomyces cerevisiae, Homo sapiens, Drosophila melanogaster and Tetrahymena thermophila.* Accession numbers are indicated for *Plasmodium falciparum* and *Toxoplasma gondii.* NI: not identified. (PDF)

**S2 Table. Oligonucleotides used in this study.**
(PDF)

**S3 Table. Minimal Data Set.** Data required to replicate all study findings reported in the article including the values used to build graphs, the values behind the means, standard deviations and statistical tests.
(XLSX)

## Acknowledgments

We are grateful to Michael Blackman for providing Compound 2. Monoclonal antibody 7.7 (α-EXP2) was obtained from The European Malaria Reagent Repository (http://www.malaria-research.eu). We thank Dave Richard (Université Laval) for providing mScarlet-Rab7 plasmid. We thank Jacobus Pharmaceuticals for WR99210 and MR4/BEI Resources, NIAID, NIH for DSM1 (MRA-1161).

## Author contributions

**Conceptualization:** Joëlle Paolo Mesén-Ramírez, Tobias Spielmann.

**Data curation:** Joëlle Paolo Mesén-Ramírez.

**Formal analysis:** Joëlle Paolo Mesén-Ramírez.

**Funding acquisition:** Tim Wolf Gilberger.

**Investigation:** Joëlle Paolo Mesén-Ramírez, Gwendolin Fuchs, Jonas Burmester, Guilherme B. Farias, Ana María Alape-Flores, Sarah Lemcke.

**Methodology:** Gwendolin Fuchs, Jonas Burmester, Guilherme B. Farias, Shamit Singla.

**Project administration:** Tim Wolf Gilberger.

**Resources:** Arne Alder, José Cubillán-Marín, Carolina Castro-Peña, Holger Sondermann, Tim Wolf Gilberger.

**Supervision:** Joëlle Paolo Mesén-Ramírez, Mónica Prado, Tobias Spielmann, Danny Wilson, Tim Wolf Gilberger.

**Validation:** Joëlle Paolo Mesén-Ramírez.

**Visualization:** Joëlle Paolo Mesén-Ramírez, Gwendolin Fuchs, Jonas Burmester, Guilherme B. Farias.

**Writing – original draft:** Joëlle Paolo Mesén-Ramírez, Danny Wilson, Tim Wolf Gilberger.

**Writing – review & editing:** Joëlle Paolo Mesén-Ramírez.

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
