## [Decision Letter · Decision Letter 0]

28 Nov 2024

PPATHOGENS-D-24-02339

HOPS/CORVET tethering complexes are critical for endocytosis and protein trafficking to invasion related organelles in malaria parasites

PLOS Pathogens

Dear Dr. Gilberger,

Thank you for submitting your manuscript to PLOS Pathogens. After careful consideration, we feel that it has merit but does not fully meet PLOS Pathogens's publication criteria as it currently stands. Therefore, we invite you to submit a revised version of the manuscript that addresses the points raised during the review process.

Please submit your revised manuscript within 60 days Jan 27 2025 11:59PM. If you will need more time than this to complete your revisions, please reply to this message or contact the journal office at plospathogens@plos.org. Please include the following items when submitting your revised manuscript:

We look forward to receiving your revised manuscript.

Kind regards,

Tracey J. Lamb

Section Editor

PLOS Pathogens

Tracey Lamb

Section Editor

PLOS Pathogens

Michael Malim

Editor-in-Chief

PLOS Pathogens

orcid.org/0000-0002-7699-2064

**Journal Requirements:**

https://journals.plos.org/plospathogens/s/submission-guidelines#loc-parts-of-a-submission

4) We have noticed that you have uploaded Supporting Information files, but you have not included a complete list of legends. Please add a full list of legends for your Supporting Information files after the references list.

5) We notice that your supplementary Figures, and Tables are included in the manuscript file. Please remove them and upload them with the file type 'Supporting Information'. Please ensure that each Supporting Information file has a legend listed in the manuscript after the references list.

Potential Copyright Issues:

i) Figure 1A. Please confirm whether you drew the images / clip-art within the figure panels by hand. If you did not draw the images, please provide (a) a link to the source of the images or icons and their license / terms of use; or (b) written permission from the copyright holder to publish the images or icons under our CC BY 4.0 license. Alternatively, you may replace the images with open source alternatives. See these open source resources you may use to replace images / clip-art:

7) We note that your Data Availability Statement is currently as follows: "All relevant data are within the manuscript and its Supporting Information files.". Please confirm at this time whether or not your submission contains all raw data required to replicate the results of your study. Authors must share the “minimal data set” for their submission. PLOS defines the minimal data set to consist of the data required to replicate all study findings reported in the article, as well as related metadata and methods (https://journals.plos.org/plosone/s/data-availability#loc-minimal-data-set-definition).

8) Please amend your detailed Financial Disclosure statement. This is published with the article. It must therefore be completed in full sentences and contain the exact wording you wish to be published.

2) State what role the funders took in the study. If the funders had no role in your study, please state: "The funders had no role in study design, data collection and analysis, decision to publish, or preparation of the manuscript.".

9) Please ensure that the funders and grant numbers match between the Financial Disclosure field and the Funding Information tab in your submission form. Note that the funders must be provided in the same order in both places as well." Currently, this funding information "DWW was funded by an Alexander Von Humboldt Fellowship, SS by an Australian Research Council RTP scholarship and DWW, TWG, SS by a DAAD/Universities Australia Collaborative Research Grant" is missing from the Funding Information tab.

Please indicate by return email the full and correct funding information for your study and confirm the order in which funding contributions should appear. Please be sure to indicate whether the funders played any role in the study design, data collection and analysis, decision to publish, or preparation of the manuscript.

**Reviewers' Comments:**

Reviewer's Responses to Questions

**Part I - Summary**

Reviewer #1: Mesén-Ramírez et al. report studies of the HOPS/CORVET tethering complexes in the malaria parasite, P. falciparum. They use the knock-sideways DNA transfection technology to tag and study VPS11, VPS16, VPS18, and VPS3 components of these complexes; similar tagging of VPS33 was unsuccessful, suggesting that that component does not tolerate the tag. VPS3 knockdown produced a more modest growth defect than seen with the other components, so it was not further characterized. Knockdown of VPS11 and VPS18 compromised both intracellular parasite maturation (through compromised endocytosis of hemoglobin and fusion to the DV) and merozoite invasion (through compromised effector protein delivery to rhoptries and micronemes). The primary effects of VPS16 knockdown related to trafficking of proteins to rhoptry and microneme organelles and defective merozoite invasion.

The studies are well-executed and involve a large number of relatively complicated DNA transfections and high-quality imaging studies.

Reviewer #2: The study by Mesen-Ramirez and colleagues investigates the function of HOPS/CORVET tethering complexes in Plasmodium falciparum, The investigators use a combination of genetic and cell biologic techniques to demonstrate as essential role for this complex during the asexual stage of parasite replication. The SLI system shows a strong growth defect for three of the HOPS/CORVET core components (VPS11,16, and 18). The use of the bloated DV assay is quite clever and demonstrates a clear disruption of the endocytic pathway for VPS11. Additional colocalization studies with well-studied cellular markers further confirms the importance of the HOPS/CORVET complex for host cell cytoplasm endocytosis. Also, in very exciting data, the team demonstrates an important role for the HOPS/CORVET complex in trafficking of materials to the rhoptries and micronemes. Overall, this is a very neat paper without major weaknesses. Some suggestions, none of which are essential, are included below.

Reviewer #3: Mesén-Ramírez and colleagues investigate the functions of conserved elements of the HOPS/CORVET complexes in vesicle trafficking events of blood-stage Plasmodium. They use a knocksideways strategy to perturb protein function and observe strong phenotypes where either early growth is disrupted and linked to incomplete delivery of endocytosed host cytosol to the digestive vacuole, or failure to generate infective merozoites where key rhoptry and microneme proteins are seen to lack typical location patterns. Overall, the work identifies elements of HOPS/CORVET core proteins in both of these processes, and discerns a difference for VPS16 that might be less involved in endocytosis. The approaches taken to verify the phenotypic observations are extensive and generally well executed. The absence of TEM-based ultrastructural analysis means that some of the conclusions of cellular defects are more indirect, but most are warranted with some caution. My main query is the interpretation of the use of mislocalistation of these proteins as equal to ‘inactivation’ or neutral loss of function, and I think that further discussion of this technology and its possible caveats should be clearly presented to the reader.

**Part II – Major Issues: Key Experiments Required for Acceptance**

Reviewer #1: The findings are in line with similar studies in other eukaryotes. Notably, they parallel studies in Toxoplasma gondii, though some modest differences are described in the Introduction. As such, the Discussion would benefit significantly from a more clear description of what findings are unique to P. falciparum and related malaria parasites. This could include insights gained about evolution of subunit sequence and structural similarities/differences in the tethering complexes.

Although the immunofluorescence and expansion microscopy images are of high quality and generally support the authors’ interpretations throughout, I wonder whether the manuscript would benefit from TEM imaging of trophozoites to show accumulation of vesicles in the cytosol that fail to fuse with the nascent DV (e.g. with and without E64 treatment) and of schizonts to establish effects on secretion to invasion-associated organelles.

Reviewer #2: These are not really "major" but included here nonetheless because they rise above typographical errors.

Major

1. For 3B, the scale of the graphs makes the phenotype difficult to appreciate. I suggest that the scale be changed to 10 and the parasites with >10 could be indicated by “pinning” to the top of the graph. For 3D, the colocalization is also difficult appreciate. I recommend including a zoomed region on in this figure to allow the better appreciation of the colocalization. If this colocalization could be quantitified, that would be an excellent addition as well.

2. For figure 5 and 6, it would be helpful to show and label examples of normal/apical and aberrant localization. For RAP1, the mislocalization is more obvious. For VPS11/RON12, it is more subtle and would benefit from some additional examples with labeling. This is an important finding and is quite subtle. I realize that this is shown better in the supplemental figure. Thus, a simple solution would be to include some of the supplemental data in the main figure itself. The data in figure 7 are very convincing. It might make the story stronger to include the respective panels in figure 5 and 6, instead of having a separate figure 7.

3. Many of the microscopy images, especially the live cell images, look like maximum projections or even widefield images of the entire parasite. I wonder if the phenotypes would be more apparent if single z-slices were shown or a subset of z-slices. This is a style comment, so it is up to the investigators

Reviewer #3: The study uses a mislocaliser with a nuclear localisation signal. I assume that this means that the mislocaliser is constantly cycling between the cytosol and nucleoplasm so that it is available to bind to its partner when the rapalog is added. So, what happens next? Is the protein of interest fully delivered to the nucleoplasm, or does it get stuck at the nucleopore complex, or does it cycle back and forth between the nucleus and cytosol as the mislocaliser is presumed to. Does it dissociate from the mislocaliser, and if so is the frequency of this different for different proteins? Would the kinetics of different cargo proteins (POIs) be different according to how they might dissociate in the nucleus? Where the site of function is known, mislocalising to the nucleus/nuclear pore complex/nuclear envelope might be sufficient to disrupt function. But if vesicle-tethering machinery is being delivered to another part of the endomembrane system (nuclear envelope here) and potentially in proximity to other compartments then could new and aberrant tethering events be promoted and more general disruption of the endomebrane ensue? No data is shown for the actual location of the bulk of the mislocalised POIs, and even so it would be hard to determine the location of minor populations. All of these uncertainties make interpretation of the mutants more complex than ‘inactivations’ as might be made for protein knockdown or knockout. Comparing the effects between proteins is potentially equally problematic here if different extents of effective ‘inactivation’ might occur. These issues need to be discussed openly and considered in the weighing up of support for the conclusions reached. I think that the approaches taken are sophisticated and address the lack of other fast mutant generators in Plasmodium, but they are not without their limitations and complexities also, perhaps especially so were the proteins of interest might be involved in many interactions within the cell. Some of the conclusions need to be treated as new untested hypotheses that warrant further investigation now.

It is a shame that TEM was not used and that the presence and appearance of membrane-bound organelles was assessed by indirect methods of protein association. If resources allowed, I’d recommend providing direct data on the presence of micronemes and host cytosol-filled vesicles.

**Part III – Minor Issues: Editorial and Data Presentation Modifications**

Reviewer #1: (No Response)

Reviewer #2: (No Response)

Reviewer #3: Minor Comments that would aid the clarity of the report and address some over-interpretations of the data presented during the narrative.

Line 107-110: In so far as the yeast/mammalian complexes are considered canonical, then defining presence and absence in apicomplexans might be useful. But this comparison doesn't allow any assessment of what might be typical of eukaryotes, including the potential ancestral state. Only two lineages are being compared here: apicomplexans against opisthokonts, so the limitations of this comparison should be acknowledged.

Line 125-6: The logic of why the function of the core HOPS/COVET subunits would be 'inactivated' by mislocalisation needs to be spelled out, including any assumptions that are being made here. Is it clear that these proteins enter the nucleoplasm where they presumablly can't function. Could some be mislocalised to nuclear pores but induce some secondary changes in HOPS/COVET activity? Would this KS proteins be expected to relocate other elements of these proteins with them? While phenotypes might be observed with this KS approach, the possible mechanics of this needs to be considered beyond 'inactivation'.

Line 138: '>97%' needs confidence values given the precision of the value stated.

Line 141: is it known if the KS approach is equally efficient for all proteins. If not, then comparing phenotypes between proteins might be difficult to do. Again, the question of how tightly KS can 'inactivate' is relevant.

Line 154: would you expect membrane vesicles to be persevered post Giemsa staining? While I agree there are white structures evident, I wonder if these can be concluded as vesicles. The ring-stage has a big whole in the middle by Giemsa, but as far as I understand this is never seen in TEM at this stage so is likely an artifact.

Line 183: I guess TEM would be required to identify these as vesicles. As seen here, they are some form of occlusion in the cytosol, but 'vesicle' is just one possible interpretation. I think the authors need to be more objective in their description of their observations and be clearer about what are then untested hypotheses.

Line 208: while vesicles might start to accumulate, their contents is speculative here and they could derive from some other process going awry here. TEM would be needed to show that they contain host cytosol.

Line 212-3: could some general perturbation of endomembrane trafficking disrupt this PI3P assignment? TEM to look at vesicle contents is still likely better.

Line 216: how is 'mostly adjacent to the DV' assessed? The DV is big, the cell is small . . .

Line 267: why was it necessary to C2 arrest the cells and then release them from this. Could natural egress not be distinguished with the build-up or not of schizonts. Could the C2 treatment cause a secondary effect? Some justification for the experimental approach is warranted at least.

Line 290: it is not very clear what this interpretation of 'closer to the PV' means seeing that the apex of the parasites is already close to the PV. How is this assessment made? What are the 'internal structures' alluded to? This description is ambiguous.

Line 291: a changed pattern of RON12 doesn't necessarily imply that it is not reaching its destination, afterall, the destination might have changed location. This is over-interpretation of the data at this point.

Line 319: is being in the cytosol and being close to the nucleus different?

It is hard to rationalise how micronemes would still form as organelles if vesicular trafficking to them is blocked. And why SP-containing proteins would then appear to accumulate in the cytosol.

PLOS authors have the option to publish the peer review history of their article (what does this mean? ). If published, this will include your full peer review and any attached files.

**Do you want your identity to be public for this peer review?** For information about this choice, including consent withdrawal, please see our Privacy Policy .

Reviewer #1: No

Reviewer #2: No

Reviewer #3: No

**Figure resubmission:**
---

## [Decision Letter · Decision Letter 1]

18 Mar 2025

Dear Dr. Gilberger,

We are pleased to inform you that your manuscript 'HOPS/CORVET tethering complexes are critical for endocytosis and protein trafficking to invasion related organelles in malaria parasites' has been provisionally accepted for publication in PLOS Pathogens.

Please also address the concern of reviewer 3 regarding wording beginning in Line 142 where inactivation may be in fact more accurately described as "functional depletion".

Best regards,

Tracey J. Lamb

Section Editor

PLOS Pathogens

Tracey Lamb

Section Editor

PLOS Pathogens

Sumita Bhaduri-McIntosh

Editor-in-Chief

PLOS Pathogens

orcid.org/0000-0003-2946-9497

Michael Malim

Editor-in-Chief

PLOS Pathogens

orcid.org/0000-0002-7699-2064

Reviewer Comments (if any, and for reference):

Reviewer's Responses to Questions

**Part I - Summary**

Reviewer #1: The authors identify and use sophisticated molecular knockdown and imaging technologies to characterize HOPS/CORVET tethering complexes in P. falciparum. These proteins serve essential roles in hemoglobin endocytosis and fusion to the DV and rhoptry/microneme formation.

Reviewer #2: As noted previously, this is an interesting manuscript that shows some novel biology in the malaria parasite. The modifications made in the revised submission (mostly) address my previous concerns.

Reviewer #3: The author's have done a good job responding to my concerns, and I'm satisfied with this. My only lingering concern is that even though they explain on Line 142 that by 'inactivation' they mean mislocalization, I don't agree this is the most appropriate term to use. The proteins are not being inactivated, rather they are depleting the opportunity for function of the protein by removing it from its expect site of action. And it is not possible to conclude that all of the protein is being effectively removed. They see clear phenotypes, but this might also be achieved with 90% mislocatalization (or either more or less). So I believe that 'functional depletion' would be a more accurate description of the understood outcome of mislocaization, and I think it would be more scientifically sound to adopt this language or something similar. But I will leave this decision in the Editor's hands.

**Part II – Major Issues: Key Experiments Required for Acceptance**

Reviewer #1: The authors have adequately addressed my questions.

Reviewer #2: None noted.

Reviewer #3: (No Response)

**Part III – Minor Issues: Editorial and Data Presentation Modifications**

Reviewer #1: (No Response)

Reviewer #2: None noted.

Reviewer #3: (No Response)

PLOS authors have the option to publish the peer review history of their article (what does this mean? ). If published, this will include your full peer review and any attached files.

**Do you want your identity to be public for this peer review?** For information about this choice, including consent withdrawal, please see our Privacy Policy .

Reviewer #1: No

Reviewer #2: No

Reviewer #3: No

---

## [Editor Report · Acceptance letter]

Dear Dr. Gilberger,

We are delighted to inform you that your manuscript, "HOPS/CORVET tethering complexes are critical for endocytosis and protein trafficking to invasion related organelles in malaria parasites," has been formally accepted for publication in PLOS Pathogens.

Best regards,

Sumita Bhaduri-McIntosh

Editor-in-Chief

PLOS Pathogens

orcid.org/0000-0003-2946-9497

Michael Malim

Editor-in-Chief

PLOS Pathogens

orcid.org/0000-0002-7699-2064